# Contextual Stochastic Bilevel Optimization

**Yifan Hu**[*]
EPFL & ETH Zürich
Switzerland

**Jie Wang**
Gatech
United States

**Yao Xie**
Gatech
United States

**Andreas Krause**
ETH Zürich
Switzerland

**Daniel Kuhn**
EPFL
Switzerland

## Abstract

We introduce *contextual stochastic bilevel optimization* (CSBO) – a stochastic bilevel optimization framework with the lower-level problem minimizing an expectation conditioned on some contextual information and the upper-level decision variable. This framework extends classical stochastic bilevel optimization when the lower-level decision maker responds optimally not only to the decision of the upper-level decision maker but also to some side information and when there are multiple or even infinite many followers. It captures important applications such as meta-learning, personalized federated learning, end-to-end learning, and Wasserstein distributionally robust optimization with side information (WDRO-SI). Due to the presence of contextual information, existing single-loop methods for classical stochastic bilevel optimization are unable to converge. To overcome this challenge, we introduce an efficient double-loop gradient method based on the Multilevel Monte-Carlo (MLMC) technique and establish its sample and computational complexities. When specialized to stochastic nonconvex optimization, our method matches existing lower bounds. For meta-learning, the complexity of our method does not depend on the number of tasks. Numerical experiments further validate our theoretical results.

## 1 Introduction

A *contextual stochastic bilevel optimization* (CSBO) problem differs from a classical stochastic bilevel optimization problem only in that its lower-level problem is conditioned on a given context $\xi$.

$$\min_{x \in \mathbb{R}^{d_x}} \quad F(x) := \mathbb{E}_{\xi \sim \mathbb{P}_\xi, \eta \sim \mathbb{P}_{\eta|\xi}}[f(x, y^*(x;\xi); \eta, \xi)] \qquad \text{(upper level)}$$

$$\text{where} \quad y^*(x;\xi) := \operatorname{argmin}_{y \in \mathbb{R}^{d_y}} \mathbb{E}_{\eta \sim \mathbb{P}_{\eta|\xi}}[g(x, y; \eta, \xi)] \quad \forall \xi \text{ and } x. \quad \text{(lower level)}$$

(1)

Here $\xi \sim \mathbb{P}_\xi$ and $\eta \sim \mathbb{P}_{\eta|\xi}$ are random vectors, with $\mathbb{P}_{\eta|\xi}$ denoting the conditional distribution of $\eta$ for a given $\xi$. The dimensions of the upper-level decision variable $x$ and the lower-level decision variable $y$ are $d_x$ and $d_y$, respectively. The functions $f$ and $g$ are continuously differentiable in $(x, y)$ for any given sample pair $(\xi, \eta)$. The function $f(x, y; \eta, \xi)$ can be nonconvex in $x$, but the function $g(x, y; \eta, \xi)$ must be strongly convex in $y$ for any given $x$, $\eta$ and $\xi$. Thus, $y^*(x;\xi)$ is the unique minimizer of the strongly convex lower-level problem for any given $x$ and $\xi$. Note that, on its own, the lower-level problem can be viewed as a contextual stochastic optimization problem [Bertsimas and Kallus, 2020] parametrized in $x$. We assume that the joint distribution of $\xi$ and $\eta$ is unknown. However, we assume that we have access to any number of independent and

---

[*]Correspondence: yifan.hu@epfl.ch.

identically distributed (i.i.d.) samples from $\mathbb{P}_\xi$, and for any given realization of $\xi$, we can generate any number of i.i.d. samples from the conditional distribution $\mathbb{P}_{\eta|\xi}$. The bilevel structure generally makes the objective function $F(x)$ nonconvex in the decision variable $x$, except for few special cases. Thus we aim to develop efficient gradient-based algorithms for finding an $\epsilon$-stationary point of the nonconvex objective function $F$, i.e., a point $\widehat{x}$ satisfying the inequality $\mathbb{E}\|\nabla F(\widehat{x})\|^2 \le \epsilon^2$.

CSBO generalizes the widely studied class of *stochastic bilevel optimization* (SBO) problems [Ghadimi and Wang, 2018] whose lower-level problem minimizes an unconditional expectation.

$$\begin{aligned}
\min_{x\in\mathbb{R}^{d_x}} \quad & \mathbb{E}_{\xi\sim\mathbb{P}_\xi}[f(x, y^*(x); \xi)] \\
\text{where} \quad & y^*(x) := \operatorname{argmin}_{y\in\mathbb{R}^{d_y}} \mathbb{E}_{\eta\sim\mathbb{P}_\eta}[g(x, y; \eta)].
\end{aligned} \tag{2}$$

Indeed, (2) is a special case of CSBO if the upper- and lower-level objective functions are stochastically independent. Another special case of CSBO is the *conditional stochastic optimization* (CSO) problem [Hu et al., 2020a,b, He and Kasiviswanathan, 2023, Goda and Kitade, 2023] representable as

$$\min_{x\in\mathbb{R}^{d_x}} \mathbb{E}_{\xi\sim\mathbb{P}_\xi}[f(x, \mathbb{E}_{\eta\sim\mathbb{P}_{\eta|\xi}}[h(x; \eta, \xi)]; \xi)]. \tag{3}$$

Indeed, (3) is a special case of CSBO if we set $g(x, y; \eta, \xi) = \|y - h(x; \eta, \xi)\|^2$, in which case the lower-level problem admits the unique closed-form solution $y^*(x, \xi) = \mathbb{E}_{\mathbb{P}_{\eta|\xi}}[h(x; \eta, \xi)]$.

**Applications.** Despite the wide applicability of SBO to various machine learning and game theory paradigms, SBO cannot capture two important cases. The first case involves the lower-level decision maker responding optimally not only to the upper-level decision $x$ but also to some side information $\xi$ like weather, spatial, and temporal information. The second case involves multiple lower-level decision makers, especially when the total number is large. CSBO well captures these two settings and encompasses various important machine learning paradigms as special cases, including meta-learning [Rajeswaran et al., 2019], personalized federated learning [Shamsian et al., 2021, Xing et al., 2022, Wang et al., 2023a], hierarchical representation learning [Yao et al., 2019], end-to-end learning [Donti et al., 2017, Sadana et al., 2023, Rychener et al., 2023, Grigas et al., 2021], Sinkhorn distributionally robust optimization (DRO) [Wang et al., 2023b], Wasserstein DRO with side information [Yang et al., 2022], information retrieval [Qiu et al., 2022], contrastive learning [Qiu et al., 2023], and instrumental variable regression [Muandet et al., 2020]. Below we provide a detailed discussion of meta-learning, personalized federated learning, and end-to-end learning.

**Meta-Learning and Personalized Federated Learning.** Both applications can be viewed as special cases of CSBO. For meta-learning with $M$ tasks or personalized federated learning with $M$ users, the goal is to find a common regularization center $\theta$ shared by all tasks or all users.

$$\min_x \ \mathbb{E}_{i\sim\mu}\mathbb{E}_{D_i^{test}\sim\rho_i} \left[l_i(y_i^*(x), D_i^{test})\right] \qquad \text{(upper level)}$$

$$\text{where } y_i^*(x) = \operatorname{argmin}_{y_i} \mathbb{E}_{D_i^{train}\sim\rho_i}\left[l_i(y_i, D_i^{train}) + \frac{\lambda}{2}\|y_i - x\|^2\right], \forall i\in[M], x. \quad \text{(lower level)}$$

$$\tag{4}$$

Here, $\mu$ is the empirical uniform distribution on $[M]$. The upper-level problem minimizes the generalization loss for all tasks/all users by tuning the joint regularization center $x$, and the lower-level problem finds an optimal regularization parameter $x_i$ close to $x$ for each individual task or user. Note that $M$ may be as large as $\mathcal{O}(10^3)$ in meta-learning and as large as $\mathcal{O}(10^9)$ in personalized federated learning. Thus, it is crucial to design methods with complexity bounds independent of $M$.

**End-to-End Learning.** Traditionally, applications from inventory control to online advertising involve a two-step approach: first estimating a demand function or the click-through rate, and then making decisions based on this estimation. End-to-end learning streamlines this into a single-step method, allowing the optimization to account for estimation errors, thereby enabling more informed decisions. This can be framed as a special case of CSBO, where the upper-level problem seeks the best estimator, while the lower-level problem makes optimal decisions based on the upper-level estimator and the contextual information $\xi$. For example, in online advertising, $x$ represents the click-through rate estimator, and $y^*(x; \xi)$ denotes the optimal advertisement display for a customer characterized by the feature vector $\xi$. For a comprehensive review, see the recent survey paper [Sadana et al., 2023].

**Challenges.** Given the wide applicability of CSBO, it is expedient to look for efficient solution algorithms. Unfortunately, when extended to CSBO, existing algorithms for SBO or CSO either

suffer from sub-optimal convergence rates or are entirely unable to handle the contextual information. Indeed, a major challenge of CSBO is to estimate $y^*(x; \xi)$ for (typically) uncountably many realizations of $\xi$. In the following, we explain in more detail why existing methods fail.

If the lower-level problem is strongly convex, then SBO can be addressed with numerous efficient single-loop algorithms [Guo and Yang, 2021, Guo et al., 2021, Chen et al., 2022a, 2021, Hong et al., 2023, Yang et al., 2021]. Indeed, as the unique minimizer $y^*(x)$ of the lower-level problem in (2) depends only on the upper-level decision variable $x$, these algorithms can sequentially update the upper- and lower-level decision variables $x$ and $y$ in a single loop while ensuring that the sequence $\{y^t\}_t$ approximates $\{y^*(x^t)\}_t$. Specifically, these approaches leverage the approximation

$$y^*(x^{t+1}) - y^*(x^t) \approx \nabla y^*(x^t)^\top (x^{t+1} - x^t),$$

which is accurate if $x$ is updated using small stepsizes. However, these algorithms generically fail to converge on CSBO problems because the minimizer $y^*(x; \xi)$ of the lower-level problem in (1) additionally depends on the context $\xi$, i.e., each realization of $\xi$ corresponds to a lower-level constraint. Consequently, there can be infinitely many lower-level constraints. It is unclear how samples from $\mathbb{P}_{\eta|\xi}$ corresponding to a fixed context $\xi$ can be reused to estimate the minimizer $y^*(x; \xi')$ corresponding to a different context $\xi'$. Since gradient-based methods sample $\xi^t$ independently in each iteration $t$, no single sequence $\{y^t\}_t$ can approximate the function $\{y^*(x^t, \xi^t)\}_t$. Guo and Yang [2021] and Hu et al. [2023] analyze a special case of the CSBO problem (1), in which $\xi$ is supported on $M$ points as shown in (4). However, the sample complexity of their algorithm grows linearly with $M$. In contrast, we develop methods for general CSBO problems and show that their sample complexities are *independent* of the support of $\xi$.

SBO problems can also be addressed with *double-loop stochastic gradient descent* (DL-SGD), which solve the lower-level problem to approximate optimality before updating the upper-level decision variable [Ji et al., 2021, Ghadimi and Wang, 2018]. We will show that these DL-SGD algorithms can be extended to CSBO problems and will analyze their sample complexity as well as their computational complexity. Unfortunately, it turns out that, when applied to CSBO problems, DL-SGD incurs high per-iteration sampling and computational costs to obtain a low-bias gradient estimator for $F$. More precisely, solving the contextual lower-level problem to $\epsilon$-optimality for a fixed $\xi$ requires $\mathcal{O}(\epsilon^{-2})$ samples from $\mathbb{P}_{\eta|\xi}$ and gradient estimators for the function $g$, which leads to a $\widetilde{\mathcal{O}}(\epsilon^{-6})$ total sample and computational complexity to obtain an $\epsilon$-stationary point of $F$.

**Methodology.** Given these observations indicating that existing methods can fail or be sub-optimal for solving the CSBO problem, we next discuss the motivation for our algorithm design. Our goal is to build gradient estimators that share the same small bias as DL-SGD but require much fewer samples and incur a much lower computational cost at the expense of a slightly increased variance.

To obtain estimators with low bias, variance, and a low sampling and computational cost, we propose here a *multilevel Monte Carlo* (MLMC) approach [Giles, 2015, Hu et al., 2021, Asi et al., 2021], which is reminiscent of the control variate technique, and combine it with inverse propensity weighting [Glynn and Quinn, 2010]. We refer to the proposed method as *random truncated MLMC* (RT-MLMC) and demonstrate that the RT-MLMC estimator for $\nabla F$ requires only $\mathcal{O}(1)$ samples from $\mathbb{P}_{\eta|\xi}$. This is a significant improvement *vis-à-vis* DL-SGD, which requires $\mathcal{O}(\epsilon^{-2})$ samples. Consequently, the sample complexity as well as the gradient complexity over $g$ (i.e., the number $g$-gradient evaluations) of RT-MLMC for finding an $\epsilon$-stationary point of $F$ is given by $\widetilde{\mathcal{O}}(\epsilon^{-4})$.

While the idea of using MLMC in stochastic optimization is not new [Hu et al., 2021, Asi et al., 2021], the construction of MLMC gradient estimators for CSBO and the analysis of the variance of the RT-MLMC gradient estimators are novel contributions of this work.

## 1.1 Our Contributions

- We introduce CSBO as a unifying framework for a broad range of machine learning tasks and optimization problems. We propose two methods, DL-SGD and RT-MLMC, and analyze their complexities; see Table 1 for a summary. When specialized to SBO and CSO problems, RT-MLMC displays the same performance as the state-of-the-art algorithms for SBO [Chen et al., 2021] and CSO [Hu et al., 2021], respectively. When specialized to classical stochastic nonconvex optimization, RT-MLMC matches the lower bounds by Arjevani et al. [2023].

Table 1: Complexity of the RT-MLMC and DL-SGD algorithms for finding an $\epsilon$-stationary point of $F$. The sample complexity refers to the total number of samples from $\mathbb{P}_\xi$ as well as $\mathbb{P}_{\eta|\xi}$.

| Nonconvex CSBO | Sample Complexity | Gradient Complexity of $g$ and $f$ | Per-iteration Memory Cost |
|---|---|---|---|
| **RT-MLMC** | $\widetilde{\mathcal{O}}(\epsilon^{-4})$ | $\widetilde{\mathcal{O}}(\epsilon^{-4}) \mid \widetilde{\mathcal{O}}(\epsilon^{-4})$ | $\mathcal{O}(d_x + d_y)$ |
| **DL-SGD** | $\widetilde{\mathcal{O}}(\epsilon^{-6})$ | $\widetilde{\mathcal{O}}(\epsilon^{-6}) \mid \widetilde{\mathcal{O}}(\epsilon^{-4})$ | $\mathcal{O}(d_x + d_y)$ |

- For meta-learning with $M$ tasks, the complexity bounds of RT-MLMC are constant in $M$. Thus, RT-MLMC outperforms the methods by Guo and Yang [2021] and Hu et al. [2023] when $M$ is large. For Wasserstein DRO with side information [Yang et al., 2022], existing methods only cater for affine and non-parametric decision rules. In contrast, RT-MLMC allows for neural network approximations. We also present the first sample and gradient complexity bounds for WDRO-SI.

- For meta-learning and Wasserstein DRO with side information, our experiments show that the RT-MLMC gradient estimator can be computed an order of magnitude faster than the DL-SGD gradient estimator, especially when the contextual lower-level problem is solved to higher accuracy.

**Preliminaries** For any function $\psi : \mathbb{R}^{d_x} \times \mathbb{R}^{d_y}$ with arguments $x \in \mathbb{R}^{d_x}$ and $y \in \mathbb{R}^{d_y}$, we use $\nabla\psi$, $\nabla_1\psi$ and $\nabla_2\psi$ to denote the gradients of $\psi$ with respect to $(x, y)$, $x$ and $y$, respectively. Similarly, we use $\nabla^2\psi$, $\nabla_{11}^2\psi$ and $\nabla_{22}^2$ to denote Hessians of $\psi$ with respect to $(x, y)$, $x$ and $y$, respectively. In addition, $\nabla_{12}^2\psi$ stands for the $(d_x \times d_y)$-matrix with entries $\partial_{x_i y_j}^2\psi$. A function $\varphi : \mathbb{R}^d \to \mathbb{R}$ is $L$-Lipschitz continuous if $|\varphi(x) - \varphi(x')| \leq L\|x - x'\|$ for all $x, x' \in \mathbb{R}^d$, and it is $S$-Lipschitz smooth if it is continuously differentiable and satisfies $\|\nabla\varphi(x) - \nabla\varphi(x')\| \leq S\|x - x'\|$ for all $x, x' \in \mathbb{R}^d$. In addition, $\varphi$ is called $\mu$-strongly convex if it is continuously differentiable and if $\varphi(x) - \varphi(x') - \nabla\varphi(x')^\top(x - x') \geq \frac{\mu}{2}\|x - x'\|^2$ for all $x, x' \in \mathbb{R}^d$. The identity matrix is denoted by $I$. Finally, we use $\widetilde{\mathcal{O}}(\cdot)$ as a variant of the classical $\mathcal{O}(\cdot)$ symbol that hides logarithmic factors.

## 2 Algorithms for Contextual Stochastic Bilevel Optimization

Throughout the paper, we make the following assumptions. Similar assumptions appear in the SBO literature [Ghadimi and Wang, 2018, Guo and Yang, 2021, Chen et al., 2022a, 2021, Hong et al., 2023].

**Assumption 2.1.** *The CSBO problem* (1) *satisfies the following regularity conditions:*

(i) *$f$ is continuously differentiable in $x$ and $y$ for any fixed $\eta$ and $\xi$, and $g$ is twice continuously differentiable in $x$ and $y$ for any fixed $\eta$ and $\xi$.*

(ii) *$g$ is $\mu_g$-strongly convex in $y$ for any fixed $x$, $\eta$ and $\xi$.*

(iii) *$f$, $g$, $\nabla f$, $\nabla g$ and $\nabla^2 g$ are $L_{f,0}$, $L_{g,0}$, $L_{f,1}$, $L_{g,1}$ and $L_{g,2}$-Lipschitz continuous in $(x, y)$ for any fixed $\eta$ and $\xi$, respectively.*

(iv) *If $(\eta, \xi) \sim \mathbb{P}_{(\eta,\xi)}$, then $\nabla f(x, y; \eta, \xi)$ is an unbiased estimator for $\nabla\mathbb{E}_{(\eta,\xi)\sim\mathbb{P}_{(\eta,\xi)}}[f(x, y; \eta, \xi)]$ with variance $\sigma_f^2$ uniformly across all $x$ and $y$. Also, if $\eta \sim \mathbb{P}_{\eta|\xi}$, then $\nabla g(x, y; \eta, \xi)$ is an unbiased estimator for $\nabla\mathbb{E}_{\eta\sim\mathbb{P}_{\eta|\xi}}[g(x, y; \eta, \xi)]$ with variance $\sigma_{g,1}^2$, and $\nabla^2 g(x, y; \eta, \xi)$ is an unbiased estimator for $\nabla^2\mathbb{E}_{\eta\sim\mathbb{P}_{\eta|\xi}}[g(x, y; \eta, \xi)]$ with variance $\sigma_{g,1}^2$ uniformly across all $x$, $y$ and $\xi$.*

Assumption 2.1 ensures that problem (1) is well-defined. In particular, by slightly adapting the proofs of [Ghadimi and Wang, 2018, Lemma 2.2] and [Hong et al., 2023, Lemma 2], it allows us to show that $F$ is $L_F$-Lipschitz continuous as well as $S_F$-Lipschitz smooth for some $L_F, S_F > 0$. Assumptions 2.1 (i-iii) also imply that the gradients of $f$ and $g$ with respect $(x, y)$ can be interchanged with the expectations with respect to $(\eta, \xi) \sim \mathbb{P}_{\eta,\xi}$ and $\eta \sim \mathbb{P}_{\eta|\xi}$. Hence, Assumptions 2.1 (i-iii) readily imply the unbiasedness of the gradient estimators imposed in Assumption 2.1 (iv). In fact, only the uniform variance bounds do not already follow from Assumptions 2.1 (i-iii).

In order to design SGD-type algorithms for problem (1), we first construct gradient estimators for $F$. To this end, we observe that the Jacobian $\nabla_1 y^*(x; \xi) \in \mathbb{R}^{d_x \times d_y}$ exists and is Lipschitz continuous in $x$ for any fixed $\xi$ thanks to [Chen et al., 2021, Lemma 2]. By the chain rule, we therefore have

$$\nabla F(x) = \mathbb{E}_{(\eta,\xi)\sim\mathbb{P}_{(\eta,\xi)}}\left[\nabla_1 f(x, y^*(x; \xi); \eta, \xi) + \nabla_1 y^*(x; \xi)^\top \nabla_2 f(x, y^*(x; \xi); \eta, \xi)\right].$$

---

**Algorithm 1** EpochSGD($K, x, \xi, y_1^0$)

---

**Input:** $\sharp$ of epochs $K$, sample $\xi$, upper-level decision $x$, initial iterate $y_1^0$.
 1: **for** $k = 1$ to $K$ **do**
 2:    **for** $j = 0$ to $2^k - 1$ **do**
 3:       Sample $\eta_k^j$ from $\mathbb{P}_{\eta|\xi}$ and update $y_k^{j+1} = y_k^j - \beta_0 2^{-k} \nabla_2 g(x, y_k^j; \eta_k^j, \xi)$.
 4:    **end for**
 5:    Update $y_{k+1}^0 = 2^{-k} \sum_{j=0}^{2^k - 1} y_k^j$.
 6: **end for**
**Output:** $y_1^0$, $y_K^0$, and $y_{K+1}^0$.

---

By following a similar procedure as in [Ghadimi and Wang, 2018], we can derive an explicit formula for $\nabla_1 y^*(x; \xi)$ (for details we refer to Appendix B) and substitute it into the above equation to obtain

$$\nabla F(x) = \mathbb{E}_{(\eta,\xi) \sim \mathbb{P}_{(\eta,\xi)}} \Big[ \nabla_1 f(x, y^*(x; \xi); \eta, \xi)$$
$$- \Big( \mathbb{E}_{\eta' \sim \mathbb{P}_{\eta|\xi}} \nabla_{12}^2 g(x, y^*(x; \xi); \eta', \xi) \Big) \Lambda(x, y^*(x; \xi); \xi) \nabla_2 f(x, y^*(x; \xi); \eta, \xi) \Big],$$

where $\Lambda(x, y; \xi) = (\mathbb{E}_{\eta \sim \mathbb{P}_{\eta|\xi}} \nabla_{22}^2 g(x, y; \eta, \xi))^{-1}$. Thus, the main challenges of constructing a gradient estimator are to compute and store the minimizer $y^*(x, \xi)$ as well as the inverse expected Hessian matrix $\Lambda(x, y; \xi)$ for all (potentially uncountably many) realizations of $\xi$. Computing these two objects *exactly* would be too expensive. In the remainder of this section, we thus derive estimators for $y^*(x; \xi)$ and $\Lambda(x, y; \xi)$, and we combine these two estimators to construct an estimator for $\nabla F(x)$.

**Estimating $y^*(x; \xi)$.** We estimate $y^*(x; \xi)$ using the gradient-based method EpochSGD by Hazan and Kale [2014], Asi et al. [2021], which involves $K$ epochs of increasing lengths. Each epoch $k = 1, \ldots, K$ starts from the average of the iterates computed in epoch $k - 1$ and then applies $2^k$ stochastic gradient steps to the lower-level problem with stepsize $2^{-k}$ (see Algorithm 1). In the following we use the output $y_{K+1}^0$ of Algorithm 1 with inputs $K$, $x$, $\xi$ and $y_0$ as an estimator for the minimizer $y^*(x; \xi)$ of the $K$-level problem. We use EpochSGD for the following two reasons. First, EpochSGD attains the optimal convergence rate for strongly convex stochastic optimization in the gradient oracle model [Hazan and Kale, 2014]. In addition, it is widely used in practical machine learning training procedures. Note that $y^*(x; \xi)$ could also be estimated via classical SGD. Even though this would lead to similar complexity results, the analysis would become more cumbersome.

**Estimating $\Lambda(x, y; \xi)$.** Following [Ghadimi and Wang, 2018], one can estimate the inverse of an expected random matrix $A$ with $0 \prec A \prec I$ using a Neumann series argument. Specifically, we have

$$[\mathbb{E}_{A \sim \mathbb{P}_A} A]^{-1} = \sum_{n'=0}^{\infty} (I - \mathbb{E}_{A \sim \mathbb{P}_A} A)^n = \sum_{n'=0}^{\infty} \prod_{n=1}^{n'} \mathbb{E}_{A_n \sim \mathbb{P}_A} (I - A_n) \approx \sum_{n'=0}^{N} \prod_{n=1}^{n'} \mathbb{E}_{A_n \sim \mathbb{P}_A} (I - A_n).$$

The truncated series on the right hand side provides a good approximation if $N \gg 1$. Assumption 2.1 (iii) implies that $0 \prec \nabla_{22}^2 g(x, y; \eta_n, \xi) \prec 2L_{g,1} I$. Hence, the above formula can be applied to $A = \frac{1}{2L_{g,1}} \nabla_{22}^2 g(x, y; \eta_n, \xi)$, which gives rise to an attractive estimator for $\Lambda(x, y; \xi)$ of the form

$$\widehat{\Lambda}(x, y; \xi) := \begin{cases} \frac{N}{2L_{g,1}} I & \text{if } \widehat{N} = 0, \\ \frac{N}{2L_{g,1}} \prod_{n=1}^{\widehat{N}} \left( I - \frac{1}{2L_{g,1}} \nabla_{22}^2 g(x, y; \eta_n, \xi) \right) & \text{if } \widehat{N} \geq 1. \end{cases} \tag{5}$$

Here, $\widehat{N}$ is a random integer drawn uniformly from $\{0, 1, \ldots, N-1\}$ that is independent of the i.i.d. samples $\eta_1, \ldots, \eta_{\widehat{N}}$ from $\mathbb{P}_{\eta|\xi}$. Chen et al. [2022b] showed that the estimator (5) displays the following properties. Its bias decays exponentially with $N$, its variance grows quadratically with $N$, and its sampling cost grows linearly with $N$. Below we call $N$ the approximation number.

**Estimating $\nabla F(x)$ via DL-SGD.** For any given $K$ and $N$, we construct the DL-SGD estimator for the gradient of $F$ by using the following procedure: (i) generate a sample $\xi$ from $\mathbb{P}_\xi$, (ii) generate i.i.d. samples $\eta'$ and $\eta''$ from the conditional distribution $\mathbb{P}_{\eta|\xi}$, (iii) run EpochSGD as described in Algorithm 1 with an arbitrary initial iterate $y_1^0$ to obtain $y_{K+1}^0$, and (iv) construct $\widehat{\Lambda}(x, y_{K+1}^0; \xi)$ as in (5). Using these ingredients, we can now construct the DL-SGD gradient estimator as

$$\widehat{v}^K(x) := \nabla_1 f(x, y_{K+1}^0; \eta'', \xi) - \nabla_{12}^2 g(x, y_{K+1}^0; \eta', \xi) \widehat{\Lambda}(x, y_{K+1}^0; \xi) \nabla_2 f(x, y_{K+1}^0; \eta'', \xi). \tag{6}$$

**Algorithm 2** SGD Framework

---

**Input:** ♯ of iterations $T$, stepsizes $\{\alpha_t\}_{t=1}^T$, initial iterate $x_1$.
1: **for** $t = 1$ to $T$ **do**
2:     Construct a gradient estimator $v(x_t)$ and update $x_{t+1} = x_t - \alpha_t v(x_t)$.
3: **end for**
**Output:** $\widehat{x}_T$ uniformly sampled from $\{x_1, ..., x_T\}$.

---

In Lemma 2 below, we will analyze the bias and variance as well as the sampling and computational costs of the DL-SGD gradient estimator. We will see that a small bias $\|\mathbb{E}[\widehat{v}^K](x) - \nabla F(x)\| \leq \epsilon$ can be ensured by setting $K = \mathcal{O}(\log(\epsilon^{-1}))$, in which case EpochSGD computes $\mathcal{O}(\epsilon^{-2})$ stochastic gradients of $g$. From now on, we refer to Algorithm 2 with $v(x) = \widehat{v}^K(x)$ as the DL-SGD algorithm.

### 2.1 RT-MLMC Gradient Estimator

The bottleneck of evaluating the DL-SGD gradient estimators is the computation of $y_{K+1}^0$. The computational costs can be reduced, however, by exploiting the telescoping sum property

$$
\widehat{v}^K(x) = \widehat{v}^1(x) + \sum_{k=1}^K [\widehat{v}^{k+1}(x) - \widehat{v}^k(x)]
$$

$$
= \widehat{v}^1(x) + \sum_{k=1}^K p_k \frac{\widehat{v}^{k+1}(x) - \widehat{v}^k(x)}{p_k} = \widehat{v}^1(x) + \mathbb{E}_{\widehat{k} \sim \mathbb{P}_{\widehat{k}}} \left[ \frac{\widehat{v}^{\widehat{k}+1}(x) - \widehat{v}^{\widehat{k}}(x)}{p_{\widehat{k}}} \right],
$$

where $\widehat{v}^{\widehat{k}}$ is defined as in (6) with $k = 1, \ldots, K$ replacing $K$, and where $\mathbb{P}_{\widehat{k}}$ is a truncated geometric distribution with $\mathbb{P}_{\widehat{k}}(\widehat{k} = k) = p_k \propto 2^{-k}$ for every $k = 1, \ldots, K$. This observation prompts us to construct the RT-MLMC gradient estimator as

$$
\widehat{v}(x) = \widehat{v}^1(x) + p_{\widehat{k}}^{-1}(\widehat{v}^{\widehat{k}+1}(x) - \widehat{v}^{\widehat{k}}(x)). \tag{7}
$$

The RT-MLMC gradient estimator has three key properties:

- It is an unbiased estimator for the DL-SGD gradient estimator, i.e., $\mathbb{E}_{\widehat{k} \sim \mathbb{P}_{\widehat{k}}}[\widehat{v}(x)] = \widehat{v}^K(x)$.

- Evaluating $\widehat{v}(x)$ requires computing $y_{k+1}^0(x, \xi)$ with probability $p_k$, which decays exponentially with $k$. To ensure a small bias, we need to set $K = \mathcal{O}(\log(\epsilon^{-1}))$, and thus $p_K = \mathcal{O}(\epsilon)$. Hence, most of the time, EpochSGD only needs to run over $k \ll K$ epochs. As a result, the average sampling and computational costs are markedly smaller for RT-MLMC than for DL-SGD.

- Since $\widehat{v}^{k+1}(x)$ and $\widehat{v}^k(x)$ differ only in $y_{k+1}^0$ and $y_k^0$, both of which are generated by EpochSGD and are thus highly correlated, $\widehat{v}^{k+1}(x) - \widehat{v}^k(x)$ has a small variance thanks to a control variate effect [Nelson, 1990]. Hence, the variance of RT-MLMC is well-controlled, as shown in Lemma 2.

In Lemma 2 below, we will analyze the bias and variance as well as the sampling and computational costs of the RT-MLMC gradient estimator. We will see that it requires only $\mathcal{O}(1)$ samples to ensure that the bias drops to $\mathcal{O}(\epsilon)$. This is in stark contrast to the DL-SGD estimator, which needs $\mathcal{O}(\epsilon^{-2})$ samples. The lower sample complexity and the corresponding lower computational cost come at the expense of an increased variance of the order $\mathcal{O}(\log(\epsilon^{-1}))$. The construction of the RT-MLMC gradient estimator is detailed in Algorithm 3. From now on, we refer to Algorithm 2 with $v(x) = \widehat{v}(x)$ as the RT-MLMC algorithm.

### 2.2 Memory and Arithmetic Operational Costs

The per-iteration memory and arithmetic operational cost of DL-SGD as well as RT-MLMC is dominated by the cost of computing the matrix-vector product

$$
\widehat{c}(x, y; \xi) := \nabla_{12}^2 g(x, y; \eta', \xi) \widehat{\Lambda}(x, y; \xi) \nabla_2 f(x, y; \eta'', \xi). \tag{8}
$$

By (5), $\widehat{\Lambda}(x, y; \xi)$ is a product of $\widehat{N}$ matrices of the form $I - 1/(2L_{g,1}) \nabla_{22}^2 g(x, y; \eta_n, \xi)$, and the $n$-th matrix coincides with the gradient of $(y - 1/(2L_{g,1}) \nabla_2 g(x, y; \eta_n, \xi))$ with respect to $y$. We can thus compute (8) recursively as follows. We first set $v = \nabla_2 f(x, y; \eta'', \xi)$. Next, we update $v$ by

---

**Algorithm 3** RT-MLMC Gradient Estimator for Conditional Bilevel Optimization

---

**Input:** Iterate $x$, largest epoch number $K$, initialization $y_1^0$, approximation number $N$

1: Sample $\xi$ from $\mathbb{P}_\xi$ and sample $\widehat{N}$ uniformly from $\{0, \ldots, N-1\}$.
2: Sample $\widehat{k}$ from the truncated geometric distribution $\mathbb{P}_{\widehat{k}}$.
3: Run EpochSGD($\widehat{k}, x, \xi, y_1^0$) and obtain $y_{\widehat{k}+1}^0$, $y_{\widehat{k}}^0$, and $y_1^0$.
4: Construct $\widehat{v}^{\widehat{k}+1}(x)$, $\widehat{v}^{\widehat{k}}(x)$, and $\widehat{v}^1(x)$ according to (6) and compute

$$\widehat{v}(x) = \widehat{v}^1(x) + p_{\widehat{k}}^{-1}(\widehat{v}^{\widehat{k}+1}(x) - \widehat{v}^{\widehat{k}}(x)).$$

**Output:** $\widehat{v}(x)$.

---

setting it to the gradient of $(y - 1/(2L_{g,1})\nabla_2 g(x, y; \eta_{\widehat{N}}, \xi))^\top v$ with respect to $y$, which is computed via automatic differentiation. This yields $v = (I - 1/(2L_{g,1})\nabla_{22}g(x, y; \eta_{\widehat{N}}, \xi))\nabla_2 f(x, y; \eta'', \xi)$. By using a backward recursion with respect to $n$, we can continue to multiply $v$ from the left with the other matrices in the expansion of $\widehat{\Lambda}(x, y; \xi)$. This procedure is highly efficient because the memory and arithmetic operational cost of computing the product of a Hessian matrix with a constant vector via automatic differentiation is bounded—up to a universal constant—by the cost of computing the gradient of the same function [Rajeswaran et al., 2019]. See Algorithm 4 in Appendix C for details. The expected arithmetic operational costs of Algorithm 4 is $\mathcal{O}(Nd)$ and the memory cost is $\mathcal{O}(d)$.

## 3 Complexity Bounds

In this section we derive the sample and gradient complexities of the proposed algorithms. We first analyze the error of the general SGD framework detailed in Algorithm 2.

**Lemma 1** (Error Analysis of Algorithm 2). *If Algorithm 2 is used to minimize an $L_\psi$-Lipschitz continuous and $S_\psi$-Lipschitz smooth fucntion $\psi(x)$ and if $\alpha \leq 1/(2S_\psi)$, then we have*

$$\mathbb{E}\|\nabla\psi(\widehat{x}_T)\|^2 \leq \frac{2A_1}{\alpha T} + \frac{2}{T}\sum_{t=1}^T \left[L_\psi\|\mathbb{E}[v(x_t) - \nabla\psi(x_t)]\| + S_\psi\alpha\mathbb{E}\|v(x_t) - \nabla\psi(x_t)\|^2\right],$$

*where $A_1 := \psi(x_1) - \min_x \psi(x)$.*

Lemma 1 sightly generalizes [Rakhlin et al., 2012, Ghadimi et al., 2016, Bottou et al., 2018]. We defer the proof to Appendix A. Thus, to prove convergence to a stationary point, we need to characterize the bias, variance, and computational costs of the DL-SGD and the RT-MLMC gradient estimators.

**Lemma 2** (Bias, Variance, Sampling Cost and Computational Cost). *We have the following results.*

* *The biases of the DL-SGD and RT-MLMC estimators match and satisfy*

$$\|\mathbb{E}\widehat{v}^K(x) - \nabla F(x)\| = \|\mathbb{E}\widehat{v}(x) - \nabla F(x)\| \leq \mu_g^{-1}(1 - \mu_g/(2L_{g,1}))^N + \mathcal{O}(N^2 2^{-K/2}),$$

  *and the corresponding variances satisfy $\mathrm{Var}(\widehat{v}^K(x)) = \mathcal{O}(N^2)$ and $\mathrm{Var}(\widehat{v}(x)) = \mathcal{O}(KN^4)$.*

* *The numbers of samples and iterations needed by EpochSGD to build a DL-SGD estimator are bounded by $N + 2^{K+1} - 1$ and $2^{K+1} - 1$, respectively. The expected numbers of samples and iterations needed for an RT-MLMC estimator are bounded by $N + 3K$ and $3K$, respectively.*

Lemma 2 implies that setting $N = \mathcal{O}(\log(\epsilon^{-1}))$ and $K = \mathcal{O}(\log(\epsilon^{-1}))$ reduces the bias to $\mathcal{O}(\epsilon)$. In this case the RT-MLMC estimators have higher variances than the DL-SGD estimators, but their variances are still of the order $\mathcal{O}(\log(\epsilon^{-1}))$. On the other hand, using RT-MLMC estimators reduces the per-iteration sampling and computational costs from $\mathcal{O}(2^K) = \mathcal{O}(\epsilon^{-2})$ to $\mathcal{O}(K) = \mathcal{O}(\log(\epsilon^{-1}))$.

Note that Hu et al. [2021] characterize the properties of general MLMC estimators and derive their complexity bounds. However, the proposed RT-MLMC estimators for CSBO problems are the first of their kind. In addition, as we need to estimate the Hessian inverse $\Lambda(x, y^*(x; \xi); \xi)$, our analysis is more involved. In contrast to Asi et al. [2021], who use MLMC techniques for estimating projections and proximal points, we use MLMC techniques for estimating gradients in bilevel optimization. The following main theorem summarizes our complexity bounds.

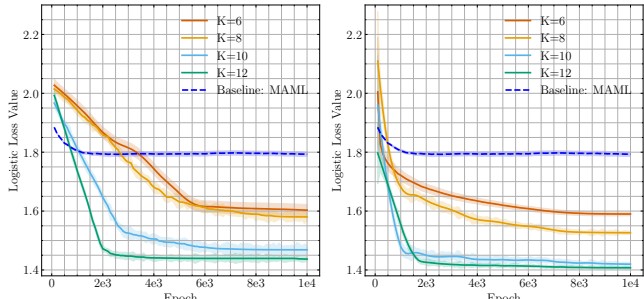

Figure 1: Test error of DL-SGD (left figure), RT-MLMC (right figure), and MAML against upper-level iterations on meta-learning. $K$ represents how accurately we solve the lower-level problem.

**Theorem 1** (Complexity Bounds). *If Assumption 2.1 holds, then Algorithm 2 based on the RT-MLMC or the DL-SGD estimator outputs an $\epsilon$-stationary point of $F$ provided that $N = \mathcal{O}(\log(\epsilon^{-1}))$, $K = \mathcal{O}(\log(\epsilon^{-1}))$, $\alpha = \mathcal{O}(\epsilon^2)$ and $T = \mathcal{O}(\epsilon^{-4})$. When using the RT-MLMC estimator, the sample complexities of $\xi$ and $\eta$ as well as the gradient complexities of $g$ and $f$ are $\widetilde{\mathcal{O}}(\epsilon^{-4})$. When using the DL-SGD estimator, the sample complexity of $\eta$ and the gradient complexity of $g$ are $\widetilde{\mathcal{O}}(\epsilon^{-6})$, while and the sample complexity of $\xi$ and the gradient complexity of $f$ are $\widetilde{\mathcal{O}}(\epsilon^{-4})$.*

**Remark.** Theorem 1 asserts that the sample complexity of $\eta$ and the gradient complexity of $g$ are much smaller for RT-MLMC than for DL-SGD, while the gradient complexities of $f$ are comparable. When specialized to SBO or CSO problems, the complexity bounds of RT-MLMC match those of the state-of-the-art algorithms ALEST for SBO problems [Chen et al., 2021] and MLMC-based methods for CSO problems [Hu et al., 2021]. When restricted to classical stochastic nonconvex optimization, the complexity bounds of RT-MLMC match the existing lower bounds [Arjevani et al., 2023]. These observations further highlight the effectiveness of RT-MLMC across various settings.

## 4 Applications and Numerical Experiments

### 4.1 Meta-Learning

Optimization-based meta-learning [Finn et al., 2017, Rajeswaran et al., 2019] aims to find a common shared regularization parameter for multiple similar yet different machine learning tasks in order to avoid overfitting when training each task separately on their datasets that each only processes a few data points. Recall Equation (4), the objective function of the optimization-based meta-learning [Rajeswaran et al., 2019],

$$\min_x \ \mathbb{E}_{i\sim\mu}\mathbb{E}_{D_i^{test}\sim\rho_i} \left[l_i(y_i^*(x), D_i^{test})\right]$$

$$\text{where } y_i^*(x) = \operatorname{argmin}_{y_i} \mathbb{E}_{D_i^{train}\sim\rho_i}\left[l_i(y_i, D_i^{train}) + \frac{\lambda}{2}\|y_i - x\|^2\right], \forall i \in [M] \text{ and } x. \tag{9}$$

where $\mu$ is the distribution over all $M$ tasks, $\rho_i$ is the distribution of data from the task $i$, $D_i^{train}$ and $D_i^{test}$ are the training and testing dataset of the task $i$, $x$ is the shared parameter of all tasks and $y_i^*(x)$ is the best parameter for a regularized objective of task $i$, $l_i$ is a loss function that measures the average loss on the dataset of the $i$-th task, and $\lambda$ is the regularization parameter to ensure the optimal solution obtained from the lower-level problem is not too far from the shared parameter obtained from the upper-level problem. Note that such a problem also occurs in personalized federated learning with each lower level being one user.

Note that the task distribution $\mu$ is usually replaced by averaging over all $M$ tasks. In such cases, existing works [Guo and Yang, 2021, Rajeswaran et al., 2019] only demonstrate a convergence rate that scales linearly with the number of tasks $M$. In contrast, the sample complexity of our proposed method does not depend on the number of tasks $M$, enabling substantially faster computation for a larger $M$. The seminal work, Model-agnostic Meta-learning (MAML) [Finn et al., 2017], is an approximation of Problem (9) via replacing $y_i^*(x)$ with one-step gradient update, i.e., $\widehat{y_i}(x) := x - \lambda^{-1}\nabla l_i(x, D_i^{train})$.

We study the case where the loss function $l_i(x, D), \forall i$ is a multi-class logistic loss using a linear classifier. The experiment is examined on tinyImageNet [Mnmoustafa, 2017] by pre-processing

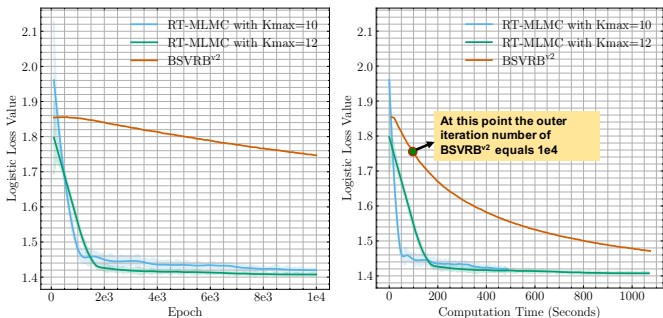

Figure 2: Left: Performance of our proposed RT-MLMC algorithm and BSVRB$^{\text{v2}}$ against upper-level iterations on meta-learning. Right: Performance of algorithms against the total computational time on meta-learning.

it using the pre-trained ResNet-18 network [He et al., 2016] to extract linear features. Since the network has learned a rich set of hierarchical features from the ImageNet dataset [Deng et al., 2009], it typically extracts useful features for other image datasets. Note that each task consists of labels of similar characteristics.

Figure 1 presents the average of logistic loss evaluated on the test dataset against the number of iterations, with each iteration representing one upper-level update. From the plot, we see that both DL-SGD and RT-MLMC methods tend to have better generalization performance when using a larger number of levels $K$. As shown in Table 2, RT-MLMC is about 9 times faster to compute the upper-level gradient estimator than DL-SGD when $K$ is large.

Table 2: The computation time of DL-SGD/RT-MLMC gradient estimators on meta-learning.

| $K$ | DL-SGD | | RT-MLMC | |
|---|---|---|---|---|
| | Mean | Variance | Mean | Variance |
| 6 | **2.65e-02** | 6.34e-03 | 2.73e-02 | 1.46e-02 |
| 8 | 7.23e-02 | 7.77e-03 | **3.41e-02** | 1.85e-02 |
| 10 | 2.48e-01 | 2.75e-02 | **4.93e-02** | 4.06e-02 |
| 12 | 9.38e-01 | 3.71e-02 | **1.08e-01** | 5.44e-02 |

In contrast, the MAML baseline does not have superior performance since the one-step gradient update does not solve the lower-level problem to approximate global optimality. In Appendix D.1, we provide numerical results for a modified MAML algorithm with multiple gradient updates, which achieves better performance compared to MAML but is still worse than our proposed method.

After the initial submission of the paper, a con-current work Hu et al. [2023] proposed two types of algorithms (BSVRB$^{\text{v1}}$ and BSVRB$^{\text{v2}}$) that apply to the meta-learning formulation (9). Their proposed algorithm BSVRB$^{\text{v1}}$ is computationally expansive as it requires the exact computation of the inverse of Hessian matrix (which is of size $5120 \times 5120$ in this example) with respect to $\theta$ in each iteration of the upper-level update. In the following, we compare the performance of our algorithm with their proposed Hessian-free algorithm BSVRB$^{\text{v2}}$ in Figure 2.

In the left plot of Figure 2, we examine the performance of RT-MLMC method and BSVRB$^{\text{v2}}$ by running the same number of total epochs. It shows that RT-MLMC method has much better performance in terms of test error. In the right plot of Figure 2, we examine the performance of RT-MLMC method and BSVRB$^{\text{v2}}$ by running the same amount of computational time. Although the per-upper-level-iteration computational costs of BSVRB$^{\text{v2}}$ is small, it takes a much longer time for BSVRB$^{\text{v2}}$ to achieve a similar test error as RT-MLMC.

### 4.2 Wasserstein DRO with Side Information

The WDRO-SI [Yang et al., 2022] studies robust stochastic optimization with side information [Bertsimas and Kallus, 2020]. Let $\xi$ denote the side information and $\eta$ denote the randomness dependent on $\xi$. The WDRO-SI seeks to find a parameterized mapping $f(x; \xi)$ from the side information $\xi$ to a decision $w$ that minimizes the expected loss w.r.t. $\eta$ under the worst-case distributional shifts over $(\xi, \eta)$. Rigorously, with a penalty on the distributional robust constraint, WDRO-SI admits the form

$$\min_x \max_{\mathbb{P}} \left\{ \mathbb{E}_{(\xi, \eta) \sim \mathbb{P}}[l(f(x, \xi); \eta)] - \lambda \mathsf{C}(\mathbb{P}, \mathbb{P}^0) \right\}, \tag{10}$$

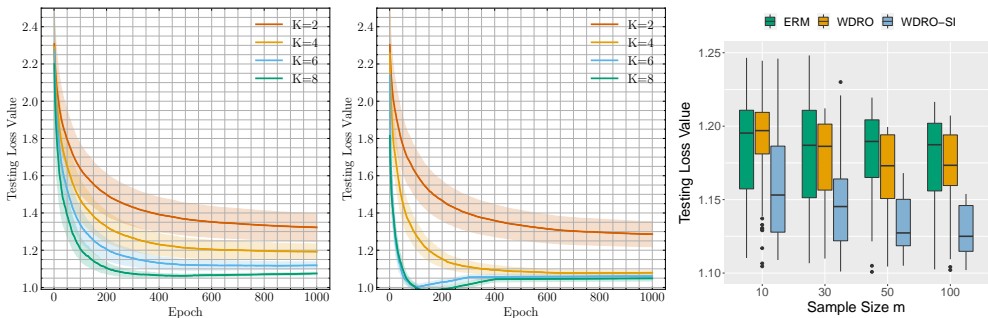

Figure 3: Test error on WDRO-SI against the number of upper-level updates for DL-SGD (left) and RT-MLMC (middle). Figure (Right) compares WDRO-SI with ERM and Wasserstein DRO that do not incorporate side information. $m$ means the number of samples of $Z$ generated from $\mathbb{P}_{Z|X}$ for each realization of $X$.

where $l(w;\eta)$ is the loss function dependent on the decision $w$ and the random variable $\eta$, $\mathbb{P}^0$ is the nominal distribution of $(\xi, \eta)$ that usually takes the form of an empirical distribution, and $\mathsf{C}(\cdot, \cdot)$ is a casual transport distance between distributions [Yang et al., 2022, Definition 1] – a variant of the Wasserstein distance that better captures the information from $\xi$. For distributionally robust feature-based newsvendor problems [Zhang et al., 2023], the covariate $\xi$ can be temporal, spatial, or weather information, $\eta$ is the random demand, $f(x;\xi)$ denotes the ordering quantity for a given $\xi$, and $l(f(x;\xi);\eta)$ characterizes the loss if the ordering quantity $f(x;\xi)$ does not match the demand $\eta$.

Incorporating the cost function of the casual transport distance used in [Yang et al., 2022] and utilizing the dual form, the WDRO-SI problem in (10) can be reformulated as a special case of CSBO:

$$\min_x \; \mathbb{E}_{\xi \sim \mathbb{P}_\xi^0} \mathbb{E}_{\eta \sim \mathbb{P}_{\eta|\xi}^0} \left[ l(f(x; y^*(x;\xi)), \eta) - \lambda \|y^*(x;\xi) - \xi\|^2 \right] \qquad \text{(upper level)}$$

$$\text{where } y^*(x;\xi) := \operatorname{argmin}_{\widetilde{\xi}} \mathbb{E}_{\eta \sim \mathbb{P}_{\eta|\xi}^0} \left[ -l(f(x;\widetilde{\xi}), \eta) + \lambda \|\xi - \widetilde{\xi}\|^2 \right], \; \forall \widetilde{\xi} \text{ and } x. \quad \text{(lower level)}$$
$$(11)$$

The original work [Yang et al., 2022] only allows affine function $f$ or non-parametric approximation, while our approach allows using neural network approximation such that $f(x;\xi)$ is a neural network parameterized by $x$. Using Theorem 1, we obtain the first sample and gradient complexities for WDRO-SI. For the distributionally robust feature-based newsvendor problems, we compare the performance of DL-SGD and RT-MLMC. We compare with ERM and WDRO, which do not incorporate side information.

Fig. 3 (left) and (middle) present the results of test loss versus the number of upper-level iterations for DL-SGD and RT-MLMC, respectively. From the plot, using a larger number of epochs $K$ for the lower-level problem generally admits lower testing loss values, i.e., better generalization performance.

Fig. 3 (right) highlights the importance of incorporating side information as the performance of WDRO-SI outperforms the other two baselines. In addition, more observations of $\eta$ for a given side information $\xi$ can enhance the performance. Table 3 reports the computational time for DL-SGD and RT-MLMC gradient estimators, and RT-MLMC is significantly faster since it properly balances the bias-variance-computation trade-off for the gradient simulation.

Table 3: Computation time of DL-SGD/RT-MLMC gradient estimators for WDRO-SI.

| $K$ | DL-SGD | | RT-MLMC | |
| --- | --- | --- | --- | --- |
| | Mean | Variance | Mean | Variance |
| 2 | 1.27e-02 | 2.67e-03 | **5.04e-03** | 7.26e-04 |
| 4 | 5.25e-02 | 2.58e-03 | **1.25e-02** | 8.26e-03 |
| 6 | 1.68e-01 | 2.74e-03 | **2.02e-02** | 9.39e-03 |
| 8 | 4.63e-01 | 2.08e-03 | **3.41e-02** | 1.68e-02 |

## 5  Conclusion

We introduced the class of contextual stochastic bilevel optimization problems, which involve a contextual stochastic optimization problem at the lower level. In addition, we designed efficient gradient-based solution schemes and analyzed their sample and gradient complexities. Numerical results on two complementary applications showcase the expressiveness of the proposed problem class as well as the efficiency of the proposed algorithms. Future research should address generalized CSBO problems with constraints at the lower level, which will require alternative gradient estimators.

## Acknowledgments and Disclosure of Funding

This research was supported by the Swiss National Science Foundation under the NCCR Automation, grant agreement 51NF40_180545, an NSF CAREER CCF-1650913, NSF DMS-2134037, CMMI-2015787, CMMI-2112533, DMS-1938106, DMS-1830210, and the Coca-Cola Foundation.

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

# Appendices

## A Proofs of Technical Results

*Proof of Lemma 1.* Since the function $\psi$ is $S_\psi$-Lipschitz smooth, we have

$$\mathbb{E}\psi(x_{t+1}) - \psi(x_t)$$

$$\leq \mathbb{E}\nabla\psi(x_t)^\top(x_{t+1} - x_t) + \frac{S_\psi}{2}\mathbb{E}\|x_{t+1} - x_t\|^2$$

$$= -\alpha_t\mathbb{E}\nabla\psi(x_t)^\top v(x_t) + \frac{S_\psi\alpha_t^2}{2}\mathbb{E}\|v(x_t)\|^2$$

$$\leq -\alpha_t\mathbb{E}\|\nabla\psi(x_t)\|^2 + \alpha_t\mathbb{E}\nabla\psi(x_t)^\top(\nabla\psi(x_t) - v(x_t))$$
$$+ S_\psi\alpha_t^2\mathbb{E}\|v(x_t) - \nabla\psi(x_t)\|^2 + S_\psi\alpha_t^2\mathbb{E}\|\nabla\psi(x_t)\|^2$$

$$= -(\alpha_t - S_\psi\alpha_t^2)\mathbb{E}\|\nabla\psi(x_t)\|^2 + \alpha_t\mathbb{E}\nabla\psi(x_t)^\top(\nabla\psi(x_t) - v(x_t)) + S_\psi\alpha_t^2\mathbb{E}\|v(x_t) - \nabla\psi(x_t)\|^2$$

$$= -(\alpha_t - S_\psi\alpha_t^2)\mathbb{E}\|\nabla\psi(x_t)\|^2 + \alpha_t\mathbb{E}[\nabla\psi(x_t)^\top\mathbb{E}[\nabla\psi(x_t) - v(x_t) \mid x_t]]$$
$$+ S_\psi\alpha_t^2\mathbb{E}\|v(x_t) - \nabla\psi(x_t)\|^2$$

$$\leq -\alpha_t/2\mathbb{E}\|\nabla\psi(x_t)\|^2 + \alpha_t\mathbb{E}[\nabla\psi(x_t)^\top\mathbb{E}[\nabla\psi(x_t) - v(x_t) \mid x_t]] + S_\psi\alpha_t^2\mathbb{E}\|v(x_t) - \nabla\psi(x_t)\|^2$$

$$\leq -\alpha_t/2\mathbb{E}\|\nabla\psi(x_t)\|^2 + \alpha_t\mathbb{E}[\|\nabla\psi(x_t)\|\|\mathbb{E}[\nabla\psi(x_t) - v(x_t) \mid x_t]\|] + S_\psi\alpha_t^2\mathbb{E}\|v(x_t) - \nabla\psi(x_t)\|^2,$$

where the first inequality uses Lipschitz smoothness of $\psi$, the first equality uses the updates of the SGD algorithm, the second inequality uses the Cauchy-Schwarz inequality, the third equality uses the conditional expectation and the tower property, the third inequality uses the fact that $\alpha_t \leq 1/(2S_\psi)$, and the fourth inequality uses the Cauchy-Schwarz inequality.

Rearranging terms and setting $\alpha_t = \alpha$, we have

$$\mathbb{E}\|\nabla\psi(x_t)\|^2 \leq \frac{2(\mathbb{E}\psi(x_t) - \mathbb{E}\psi(x_{t+1}))}{\alpha}$$
$$+ 2\mathbb{E}\|\nabla\psi(x_t)\|\|\mathbb{E}[\nabla\psi(x_t) - v(x_t) \mid x_t]\| + 2S_\psi\alpha\mathbb{E}\|v(x_t) - \nabla\psi(x_t)\|^2.$$

Averaging from $t = 1$ to $t = T$, we have

$$\mathbb{E}\|\nabla\psi(\widehat{x}_T)\|^2 = \frac{1}{T}\sum_{t=1}^{T}\mathbb{E}\|\nabla\psi(x_t)\|^2 \leq \frac{2(\psi(x_1) - \min_x\psi(x))}{\alpha T}$$

$$+ \frac{2}{T}\sum_{t=1}^{T}\mathbb{E}\|\nabla\psi(x_t)\|\|\mathbb{E}[\nabla\psi(x_t) - v(x_t) \mid x_t]\| + \frac{2S_\psi\alpha}{T}\sum_{t=1}^{T}\mathbb{E}\|v(x_t) - \nabla\psi(x_t)\|^2$$

$$\leq \frac{2(\psi(x_1) - \min_x\psi(x))}{\alpha T} + \frac{2}{T}\sum_{t=1}^{T}\Big[L_\psi\|\mathbb{E}[\nabla\psi(x_t) - v(x_t)]\| + S_\psi\alpha\mathbb{E}\|v(x_t) - \nabla\psi(x_t)\|^2\Big],$$

where the inequality holds as $\psi$ is $L_\psi$-Lipschitz continuous and thus $\|\nabla\psi(x)\| \leq L_\psi$ for all $x$. $\qquad\square$

To demonstrate the bias, variance as well as sampling and computational costs for building DL-SGD and RT-MLMC gradient estimators, we first show the iterate convergence of EpochSGD on the lower-level minimization problem with the side information. The analysis follows similarly as [Hazan and Kale, 2014, Asi et al., 2021].

**Lemma 3** (Error of EpochSGD). *For given $x$ and $\xi$, the iterates $y_{K+1}^0$ of Algorithm 1 with $\beta_0 = (4\mu_g)^{-1}$ satisfies*

$$\mathbb{E}\|y_{K+1}^0 - y^*(x;\xi)\|^2 \leq 2L_{g,0}^2\mu_g^{-2}2^{-(K+1)}.$$

It is important to note that the initial stepsize, denoted as $\beta_0$, doesn't necessarily have to be equal to $(4\mu_g)^{-1}$. Indeed, equivalent results can be achieved with a constant $\beta_0 > 0$. The choice of this

specific $\beta_0$ value is primarily to streamline the analysis. In reality, there are numerous instances where the value of $\mu_g$ is unknown beforehand. Under such circumstances, one practical approach could be to set $\beta_0$ to a standard value, such as $0.4$.

*Proof of Lemma 3.* Denote $G(x, y; \xi) := \mathbb{E}_{\eta|\xi} g(x, y; \eta, \xi)$. For the ease of notation, throughout the proof, $\mathbb{E}$ denotes taking full expectation conditioned on a given $x$ and $\xi$. Utilizing the update of $y$ in the $k$-th epoch of EpochSGD algorithm, it holds for any $y$ and any $j = 0, \ldots, 2^k - 1$ that

$$
\begin{aligned}
&\mathbb{E}\|y_k^{j+1} - y\|^2 \\
=&\mathbb{E}\|y_k^j - \beta_k \nabla_2 g(x, y_k^j; \eta_k^j, \xi) - y\|^2 \\
=&\mathbb{E}\|y_k^j - y\|^2 + \beta_k^2 \mathbb{E}\|\nabla_2 g(x, y_k^j; \eta_k^j, \xi)\|^2 - 2\beta_k \mathbb{E} \nabla_2 g(x, y_k^j; \eta_k^j, \xi)^\top (y_k^j - y) \\
\leq&\mathbb{E}\|y_k^j - y\|^2 + \beta_k^2 L_{g,0}^2 - 2\beta_k (\mathbb{E} G(x, y_k^j; \xi) - G(x, y; \xi)),
\end{aligned}
$$

where the inequality utilizes the convexity of $g$ and $G$ in $y$. Rearranging terms, the above inequality yields

$$
\mathbb{E} G(x, y_k^j; \xi) - G(x, y; \xi) \leq \frac{\mathbb{E}\|y_k^j - y\|^2 - \mathbb{E}\|y_k^{j+1} - y\|^2}{2\beta_k} + \frac{\beta_k L_{g,0}^2}{2}.
$$

Summing up from $j = 0$ to $j = 2^k - 1$ and dividing $2^k$ on both sides, we obtain the relation

$$
\begin{aligned}
&\mathbb{E} G(x, y_{k+1}^0; \xi) - G(x, y; \xi) \\
\leq&\frac{1}{2^k} \sum_{j=0}^{2^k - 1} \mathbb{E} G(x, y_k^j; \xi) - G(x, y; \xi) \leq \frac{\mathbb{E}\|y_k^0 - y\|^2}{2\beta_k 2^k} + \frac{\beta_k L_{g,0}^2}{2},
\end{aligned} \tag{12}
$$

where the inequality uses the convexity of $G$ in $y$ and Jensen's inequality.

Since $G(x, y; \xi)$ is $\mu_g$-strongly convex in $y$ for any given $x$ and $\xi$, it holds that

$$
\begin{aligned}
&G(x, y_1^0; \xi) - G(x, y^*(x; \xi); \xi) \\
\leq& - \nabla_2 G(x, y_1^0; \xi)^\top (y^*(x; \xi) - y_1^0) - \frac{\mu_g}{2}\|y^*(x; \xi) - y_1^0\|^2 \\
\leq& \max_y \left\{ - \nabla_2 G(x, y_1^0; \xi)^\top (y - y_1^0) - \frac{\mu_g}{2}\|y - y_1^0\|^2 \right\} \\
=&\frac{\|\nabla_2 G(x, y_1^0; \xi)\|^2}{2\mu_g} \leq \frac{L_{g,0}^2}{2\mu_g}.
\end{aligned}
$$

Next, we use induction on $k$ to show

$$
\mathbb{E} G(x, y_k^0; \xi) - G(x, y^*(x; \xi); \xi) \leq \frac{L_{g,0}^2}{2^k \mu_g}.
$$

The base step for $k = 1$ follows from the inequality established above. As for the induction step, suppose that

$$
\mathbb{E} G(x, y_k^0; \xi) - G(x, y^*(x; \xi); \xi) \leq \frac{L_{g,0}^2}{2^k \mu_g}
$$

holds for some $k \geq n$. Plugging $y = y^*(x; \xi)$ into (12), we thus find

$$
\begin{aligned}
\mathbb{E}G(x, y_{k+1}^0; \xi) - G(x, y^*(x; \xi); \xi) &\leq \frac{\mathbb{E}\|y_k^0 - y^*(x; \xi)\|^2}{2\beta_k 2^k} + \frac{\beta_k L_{g,0}^2}{2} \\
&\leq \frac{\mathbb{E}G(x, y_k^0; \xi) - G(x, y^*(x; \xi); \xi)}{\mu_g \beta_k 2^k} + \frac{\beta_0 L_{g,0}^2}{2^{K+1}} \\
&= \frac{\mathbb{E}G(x, y_k^0; \xi) - G(x, y^*(x; \xi); \xi)}{\mu_g \beta_0} + \frac{\beta_0 L_{g,0}^2}{2^{K+1}} \\
&\leq \frac{L_{g,0}^2}{\mu_g^2 \beta_0 2^k} + \frac{\beta_0 L_{g,0}^2}{2^{k+1}} \\
&= \frac{L_{g,0}^2}{\mu_g 2^{k+2}} + \frac{L_{g,0}^2}{\mu_g 2^{k+3}} \\
&\leq \frac{L_{g,0}^2}{\mu_g 2^{k+1}},
\end{aligned}
$$

where the second inequality uses the strong convexity of $G$ in $y$ and the fact that $y^*(x; \xi)$ minimizes $G$, the first equality uses our assumption that $\beta_k = \beta_0/2^k$, the third inequality uses the induction condition, and the second equality inequality uses the definition of $\beta_0 = (4\mu_g)^{-1}$. It concludes the induction. Therefore, we have

$$
\mathbb{E}G(x, y_{K+1}^0; \xi) - G(x, y^*(x; \xi); \xi) \leq \frac{L_{g,0}^2}{\mu_g 2^{K+1}}.
$$

By the $\mu_g$-strong convexity of $G(x, y, ; \xi)$ and the fact that $y^*(x; \xi)$ is the minimizer, we thus have

$$
\mathbb{E}\|y_{K+1}^0 - y^*(x; \xi)\|^2 \leq \frac{2}{\mu_g}\mathbb{E}G(x, y_{K+1}^0; \xi) - G(x, y^*(x; \xi); \xi) \leq \frac{L_{g,0}^2}{\mu_g^2 2^K}.
$$

$\square$

*Proof of Lemma 2.* We first demonstrate the properties of the RT-MLMC gradient estimator and then show that of the DL-SGD gradient estimator. To facilitate the analysis, we define

$$
\begin{aligned}
V(x) = \mathbb{E}_{\mathbb{P}_\xi, \mathbb{P}_{\eta|\xi}}\Big[&\nabla_1 f(x, y^*(x; \xi); \eta, \xi) \\
&- \Big(\mathbb{E}_{\mathbb{P}_{\eta'|\xi}}\nabla_{12}^2 g(x, y^*(x; \xi); \eta', \xi)\Big)\mathbb{E}_{\{\eta_n\}_{n=1}^{\widehat{N}} \sim \mathbb{P}_{\eta|\xi}}[\widehat{\Lambda}(x, y^*(x; \xi); \xi)]\nabla_2 f(x, y^*(x; \xi); \eta, \xi)\Big].
\end{aligned}
$$

**RT-MLMC gradient estimator**  By the triangle inequality, we have

$$
\|\mathbb{E}\widehat{v}(x) - \nabla F(x)\| \leq \|\mathbb{E}\widehat{v}(x) - V(x)\| + \|V(x) - \nabla F(x)\|.
$$

From Chen et al. [2022b, Lemma 2.2], we know for given $x$, $y$, and $\xi$ that

$$
\|\mathbb{E}_{\{\eta_n\}_{n=1}^{\widehat{N}} \sim \mathbb{P}_{\eta|\xi}}[\widehat{\Lambda}(x, y; \xi)] - \Lambda(x, y; \xi)\| \leq \frac{1}{\mu_g}\Big(1 - \frac{\mu_g}{2L_{g,1}}\Big)^N
$$

Utilizing Lipschitz continuity of $\nabla g$ and $f$, we know that

$$
\|V(x) - \nabla F(x)\| \leq \frac{L_{g,1}L_{f,0}}{\mu_g}\Big(1 - \frac{\mu_g}{2L_{g,1}}\Big)^N.
$$

On the other hand, by the definition of $\widehat{v}(x)$, we have

$$
\begin{aligned}
&\mathbb{E}\widehat{v}(x) - V(x) \\
=&\mathbb{E}\nabla_1 f(x, y_{K+1}^0; \eta, \xi) - \mathbb{E}\nabla_1 f(x, y^*(x; \xi); \eta, \xi) \\
&+ \mathbb{E}\nabla_{12}^2 g(x, y_{K+1}^0; \eta, \xi)\Big[\widehat{\Lambda}(x, y_{K+1}^0; \xi)\Big]\nabla_2 f(x, y_{K+1}^0; \eta, \xi) \\
&- \mathbb{E}\nabla_{12}^2 g(x, y^*(x; \xi); \eta', \xi)\Big[\widehat{\Lambda}(x, y^*(x; \xi); \xi)\Big]\nabla_2 f(x, y^*(x; \xi); \eta', \xi).
\end{aligned}
$$

By the Lipschitz continuity of $\nabla f$ and Lemma 3, we have

$$\left\|\mathbb{E}\nabla_1 f(x, y_{K+1}^0; \eta, \xi) - \mathbb{E}\nabla_1 f(x, y^*(x; \xi); \eta, \xi)\right\|$$
$$\leq L_{f,1}\mathbb{E}\|y_{K+1}^0 - y^*(x; \xi)\|$$
$$\leq \frac{L_{f,1}L_{g,0}}{\mu_g 2^{K/2}}.$$

Utilizing the Lipschitz continuity of $\nabla f$, $\nabla g$, and $\nabla^2 g$ in $y$, we have

$$\|\mathbb{E}\widehat{v}(x) - V(x)\|$$
$$\leq L_{f,1}\mathbb{E}\|y_{K+1}^0 - y^*(x; \xi)\| + \frac{L_{g,1}L_{f,1}N}{L_{g,1}}\mathbb{E}\|y_{K+1}^0 - y^*(x; \xi)\| + \frac{L_{f,0}L_{g,2}N}{L_{g,1}}\mathbb{E}\|y_{K+1}^0 - y^*(x; \xi)\|$$

$$+ L_{f,0}L_{g,1}\frac{N}{L_{g,1}} \sum_{N'=1}^{N} \frac{1}{N'}N'\mathbb{E}\|y_{K+1}^0 - y^*(x; \xi)\|$$
$$\leq \frac{L_{f,1}L_{g,0}}{\mu_g 2^{K/2}} + \Big(\frac{L_{g,1}L_{f,1}N}{L_{g,1}} + \frac{L_{f,0}L_{g,2}N}{L_{g,1}} + L_{f,0}L_{g,1}\frac{N^2}{L_{g,1}}\Big)\frac{L_{g,0}}{\mu_g 2^{K/2}}$$
$$= \mathcal{O}\Big(\frac{N^2}{2^{K/2}}\Big).$$

Next, we show the sampling and computational costs. To build up the RT-MLMC gradient estimator $\widehat{v}(x)$, we need one sample of $\xi$, and the number of samples of $\eta$ from $\mathbb{P}_{\eta|\xi}$ is

$$\sum_{N'=1}^{N} \frac{1}{N'}N' + \sum_{k=1}^{K}(2^{k+1} - 1)\frac{2^{-k}}{1 - 2^{-K-1}} < N + 3K.$$

On average, the iteration needed for EpochSGD is

$$\sum_{k=1}^{K}(2^{k+1} - 1)\frac{2^{-k}}{1 - 2^{-K-1}} < 3K.$$

Next, we demonstrate the variance of $\widehat{v}(x)$. Denote

$$H_K(1) := \nabla_1 f(x, y_K^0; \eta'', \xi),$$
$$H_K(2) := \nabla_{12}^2 g(x, y_K^0; \eta', \xi)\Big[\widehat{\Lambda}(x, y_K^0; \xi)\Big]\nabla_2 f(x, y_K^0; \eta'', \xi),$$
$$H^*(1) := \nabla_1 f(x, y^*(x; \xi); \eta'', \xi),$$
$$H^*(2) := \nabla_{12}^2 g(x, y^*(x; \xi); \eta', \xi)\Big[\widehat{\Lambda}(x, y^*(x; \xi); \xi)\Big]\nabla_2 f(x, y^*(x; \xi); \eta'', \xi).$$

Thus one may rewrite $\widehat{v}(x)$ and $V(x)$ as

$$\widehat{v}(x) = \frac{H_{k+1}(1) - H_k(1) - H_{k+1}(2) + H_k(2)}{p_k} + H_1(1) - H_1(2),$$
$$V(x) = \mathbb{E}[H^*(1) - H^*(2)].$$

It holds that

$$\mathbb{E}\|\widehat{v}(x) - \mathbb{E}\widehat{v}(x)\|^2$$
$$\leq \mathbb{E}\|\widehat{v}(x) - V(x)\|^2$$
$$\leq 2\mathbb{E}\|\widehat{v}(x) - H_1(1) + H_1(2)\|^2 + 2\mathbb{E}\|H_1(1) - H_1(2) - V(x)\|^2$$
$$\leq 4\mathbb{E}\left\|\frac{1}{p_k}(H_{k+1}(1) - H_k(1))\right\|^2 + 4\mathbb{E}\left\|\frac{1}{p_k}(H_{k+1}(2) - H_k(2))\right\|^2 + 2\mathbb{E}\|H_1(1) - H_1(2) - V(x)\|^2,$$

where the first inequality holds by the definition of variance, the second inequality holds by the Cauchy-Schwarz inequality, the third inequality uses the Cauchy-Schwarz inequality and the definition of $\widehat{v}(x)$. It remains to analyze $\mathbb{E}\left\|\frac{1}{p_k}(H_{k+1}(1) - H_k(1))\right\|^2$, $\mathbb{E}\left\|\frac{1}{p_k}(H_{k+1}(2) - H_k(2))\right\|^2$, and $\mathbb{E}\|H_1(1) - H_1(2) - V(x)\|^2$.

- For the first term, we have

$$\mathbb{E}\Big\|\frac{1}{p_k}(H_{k+1}(1)-H_k(1))\Big\|^2$$

$$=\sum_{k=1}^{K}p_k^{-1}\mathbb{E}\|H_{k+1}(1)-H_k(1)\|^2$$

$$\leq\sum_{k=1}^{K}p_k^{-1}L_{f,1}^2\mathbb{E}\|y_{k+1}^0-y_k^0\|^2$$

$$\leq\sum_{k=1}^{K}p_k^{-1}L_{f,1}^2 2(\mathbb{E}\|y_{k+1}^0-y^*(x;\xi)\|+\mathbb{E}\|y^*(x;\xi)-y_k^0\|^2)$$

$$\leq\sum_{k=1}^{K}p_k^{-1}L_{f,1}^2\frac{6L_{g,0}^2}{\mu_g^2 2^k}$$

$$\leq KL_{f,1}^2\frac{6L_{g,0}^2}{\mu_g^2},$$

where the first inequality uses the Lipschitz continuity of $\nabla f$, the second inequality uses the Cauchy-Schwarz inequality, the third inequality uses Lemma 3, and the last inequality uses the definition of $p_k$.

- For the second term, we may conduct a similar analysis.

$$\mathbb{E}\Big\|\frac{1}{p_k}(H_{k+1}(2)-H_k(2))\Big\|^2$$

$$=\sum_{k=1}^{K}p_k^{-1}\mathbb{E}\|H_{k+1}(2)-H_K(2)\|^2$$

$$\leq\sum_{k=1}^{K}p_k^{-1}6\Big(L_{f,1}^2 N^2+\frac{L_{f,0}^2 L_{g,2}^2 N^2}{L_{g,1}^2}+L_{f,0}^2 N^4\Big)\mathbb{E}\|y_{K+1}^0-y_K^0\|^2$$

$$\leq\sum_{k=1}^{K}p_k^{-1}2\Big(\frac{L_{g,0}^2}{\mu_g^2 2^K}+\frac{L_{g,0}^2}{\mu_g^2 2^{K-1}}\Big)6\Big(L_{f,1}^2 N^2+\frac{L_{f,0}^2 L_{g,2}^2 N^2}{L_{g,1}^2}+L_{f,0}^2 N^4\Big)$$

$$=\frac{36KL_{g,0}^2}{\mu_g^2}\Big(L_{f,1}^2 N^2+\frac{L_{f,0}^2 L_{g,2}^2 N^2}{L_{g,1}^2}+L_{f,0}^2 N^4\Big),$$

where the first inequality follows utilizing Lipschitz continuity of $f$, $\nabla f$, $\nabla g$, $\nabla^2 g$ in $y$, the second inequality uses Lemma 3, and the last inequality uses the definition of $p_k$.

- For the third term, we have

$$\mathbb{E}\|H_1(1)-H_1(2)-V(x)\|^2$$

$$\leq 2\mathbb{E}\|H_1(1)-\mathbb{E}H^*(1)\|^2+2\mathbb{E}\|H_1(2)-\mathbb{E}H^*(2)\|^2$$

Notice that

$$\mathbb{E}\|H_1(1)-\mathbb{E}H^*(1)\|^2$$

$$=\mathbb{E}\|\nabla_1 f(x,y_1^0;\eta,\xi)-\mathbb{E}\nabla_1 f(x,y_1^0;\eta,\xi)\|^2$$

$$\qquad+\mathbb{E}\|\mathbb{E}\nabla_1 f(x,y_1^0;\eta,\xi)-\mathbb{E}\nabla_1 f(x,y^*(x;\xi);\eta,\xi)\|^2$$

$$\leq\sigma_f^2+\frac{L_{f,1}^2 L_{g,0}^2}{2\mu_g^2},$$

where the first equality holds as $\mathbb{E}\|a+b\|^2 = \mathbb{E}\|a\|^2 + \mathbb{E}\|b\|^2 + 2\mathbb{E}a^\top b$, and last inequality holds by Lemma 3 and the Lipschitz continuity of $\nabla f$. On the other hand, we have

$$
\begin{aligned}
&\mathbb{E}\|H_1(2) - \mathbb{E}H^*(2)\|^2 \\
=&\mathbb{E}\|H_1(2) - \mathbb{E}H_1(2)\|^2 + \mathbb{E}\|\mathbb{E}H_1(2) - \mathbb{E}H^*(2)\|^2 \\
\leq&\mathbb{E}\|H_1(2)\|^2 + \mathbb{E}\|\mathbb{E}H_1(2) - \mathbb{E}H^*(2)\|^2 \\
\leq& L_{g,1}^2 L_{f,0}^2 \sum_{\widehat{N}=1}^{N} \frac{1}{\widehat{N}} \frac{N^2}{4L_{g,1}} + 6\Big(L_{f,1}^2 N^2 + \frac{L_{f,0}^2 L_{g,2}^2 N^2}{L_{g,1}^2} + L_{f,0}^2 N^4\Big) \frac{L_{g,0}^2}{2\mu_g^2} \\
=& L_{g,1}^2 L_{f,0}^2 \frac{N^2}{4L_{g,1}} + 6\Big(L_{f,1}^2 N^2 + \frac{L_{f,0}^2 L_{g,2}^2 N^2}{L_{g,1}^2} + L_{f,0}^2 N^4\Big) \frac{L_{g,0}^2}{2\mu_g^2},
\end{aligned}
$$

where the first equality uses $\mathbb{E}\|a+b\|^2 = \mathbb{E}\|a\|^2 + \mathbb{E}\|b\|^2 + 2\mathbb{E}a^\top b$, and last inequality holds by Lemma 3 and the Lipschitz continuity.

As a result, we conclude that the variance satisfies that

$$
\mathbb{E}\|\widehat{v}(x) - \mathbb{E}\widehat{v}(x)\|^2 \leq \mathbb{E}\|\widehat{v}(x) - V(x)\|^2 = \mathcal{O}(KN^4).
$$

**DL-SGD gradient estimators**  Note that $\mathbb{E}\widehat{v}(x) = \mathbb{E}\widehat{v}^K(x)$. Thus the bias follows directly from the analysis of RT-MLMC.

Next, we consider the per-iteration sampling costs and the average number of iterations for the EpochSGD. DL-SGD runs EpochSGD as a subroutine for $2^{K+1} - 1$ number of iterations. Thus the sampling cost on average is $N + 2^{K+1} - 1$.

Consider the variance of the DL-SGD method. Note that

$$
\widehat{v}^K(x) = H_{K+1}(1) - H_{K+1}(2)
$$

Following a similar decomposition as we did for the third term in bounding the variance of RT-MLMC estimators, we have

$$
\begin{aligned}
&\mathbb{E}\|\widehat{v}^K(x) - \mathbb{E}\widehat{v}^K(x)\|^2 \\
\leq&\mathbb{E}\|\widehat{v}^K(x) - V(x)\|^2 \\
\leq&2\mathbb{E}\|H_{K+1}(1) - \mathbb{E}H^*(1)\|^2 + 2\mathbb{E}\|H_{K+1}(2) - \mathbb{E}H^*(2)\|^2 \\
\leq&2\mathbb{E}\|H_{K+1}(1) - \mathbb{E}H_{K+1}(1)\|^2 + 2\mathbb{E}\|\mathbb{E}H_{K+1}(1) - \mathbb{E}H^*(1)\|^2 + 2\mathbb{E}\|H_{K+1}(2)\|^2 \\
&\quad + 2\mathbb{E}\|\mathbb{E}H_{K+1}(2) - \mathbb{E}H^*(2)\|^2 \\
\leq&2\sigma_f^2 + \frac{L_{f,1}^2 L_{g,0}^2}{2^K \mu_g^2} + L_{g,1}^2 L_{f,0}^2 \frac{N^2}{2L_{g,1}} + 12\Big(L_{f,1}^2 N^2 + \frac{L_{f,0}^2 L_{g,2}^2 N^2}{L_{g,1}^2} + L_{f,0}^2 N^4\Big) \frac{L_{g,0}^2}{2^{K+1}\mu_g^2} \\
=&\mathcal{O}(N^2 + N^4 2^{-K}),
\end{aligned}
$$

where the first inequality uses the definition of variance, the second inequality uses the Cauchy-Schwarz inequality, and the third and the last equality uses Lipschitz continuity of $f$, $\nabla f$ and $\nabla g$ and follows a similar argument as in bounding the third term for the variance of RT-MLMC estimators. Note that to control the bias, we let both $N$ and $K$ to be of order $\mathcal{O}(\log(\epsilon^{-1}))$. Thus $N^2$ is the dominating term in the variance of DL-SGD gradient estimator. □

Next, we demonstrate the proof of Theorem 1.

*Proof of Theorem 1.* We first demonstrate the analysis for RT-MLMC.

**RT-MLMC gradient method:** combining Lemmas 1 and 2, we know that

$$\mathbb{E}\|\nabla F(\widehat{x}_T)\|^2$$

$$\leq \frac{2(\mathbb{E}F(x_1) - \min_x F(x))}{\alpha T} + \frac{2}{T}\sum_{t=1}^{T}\mathbb{E}\|\nabla F(x_t)\|\|\mathbb{E}[\nabla F(x_t) - \widehat{v}(x_t) \mid x_t]\|$$

$$+ \frac{2S_F\alpha}{T}\sum_{t=1}^{T}\mathbb{E}\|\widehat{v}(x_t) - \nabla F(x_t)\|^2$$

$$\leq \frac{2(\mathbb{E}F(x_1) - \min_x F(x))}{\alpha T} + \frac{2}{T}\sum_{t=1}^{T}\mathbb{E}\|\nabla F(x_t)\|\|\mathbb{E}[\nabla F(x_t) - \widehat{v}(x_t) \mid x_t]\|$$

$$+ \frac{4S_F\alpha}{T}\sum_{t=1}^{T}\mathbb{E}\|\widehat{v}(x_t) - V(x_t)\|^2 + \frac{4S_F\alpha}{T}\sum_{t=1}^{T}\mathbb{E}\|V(x_t) - \nabla F(x_t)\|^2$$

$$= \mathcal{O}\Big(\frac{1}{\alpha T} + S_F\frac{1}{\mu_g}\Big(1 - \frac{\mu_g}{2L_{g,1}}\Big)^N + S_F\frac{N^2}{\mu_g 2^{K/2}} + \alpha S_F\frac{KN^4}{\mu_g^2} + \alpha\frac{1}{\mu_g^2}\Big(1 - \frac{\mu_g}{2L_{g,1}}\Big)^{2N}\Big).$$

To ensure that $\widehat{x}_T$ is an $\epsilon$-stationarity point, it suffices to let the right hand side of the inequality above to be $\mathcal{O}(\epsilon^2)$. Correspondingly, we set $N = \mathcal{O}(\log(\epsilon^{-1}))$, $K = \mathcal{O}(\log(\epsilon^{-2}))$, $\alpha = \mathcal{O}(T^{-1/2})$, and $T = \widetilde{\mathcal{O}}(\epsilon^{-4})$. As a result, the total sampling and the gradient complexity over $g$ are of order $3KT = \widetilde{\mathcal{O}}(\epsilon^{-4})$. Since at each upper iteration, we only compute one gradient of $f$, the gradient complexity over $f$ is $T = \widetilde{\mathcal{O}}(\epsilon^{-4})$.

Next, we demonstrate the analysis for DL-SGD.

**DL-SGD gradient method:** combining Lemmas 1 and 2, we have

$$\mathbb{E}\|\nabla F(\widehat{x}_T)\|^2$$

$$\leq \frac{2(\mathbb{E}F(x_1) - \min_x F(x))}{\alpha T} + \frac{2}{T}\sum_{t=1}^{T}\mathbb{E}\|\nabla F(x_t)\|\|\mathbb{E}[\nabla F(x_t) - \widehat{v}^K(x_t) \mid x_t]\|$$

$$+ \frac{2S_F\alpha}{T}\sum_{t=1}^{T}\mathbb{E}\|\widehat{v}^K(x_t) - \nabla F(x_t)\|^2$$

$$\leq \frac{2(\mathbb{E}F(x_1) - \min_x F(x))}{\alpha T} + \frac{2}{T}\sum_{t=1}^{T}\mathbb{E}\|\nabla F(x_t)\|\|\mathbb{E}[\nabla F(x_t) - \widehat{v}^K(x_t) \mid x_t]\|$$

$$+ \frac{4S_F\alpha}{T}\sum_{t=1}^{T}\mathbb{E}\|\widehat{v}^K(x_t) - V(x_t)\|^2 + \frac{4S_F\alpha}{T}\sum_{t=1}^{T}\mathbb{E}\|V(x_t) - \nabla F(x_t)\|^2$$

$$= \mathcal{O}\Big(\frac{1}{\alpha T} + S_F\frac{1}{\mu_g}\Big(1 - \frac{\mu_g}{2L_{g,1}}\Big)^N + S_F\frac{N^2}{\mu_g 2^{K/2}} + \alpha S_F N^2 + \alpha\frac{1}{\mu_g^2}\Big(1 - \frac{\mu_g}{2L_{g,1}}\Big)^{2N}\Big).$$

To ensure that $\widehat{x}_T$ is an $\epsilon$-stationarity point, it suffices to let $N = \mathcal{O}(\log(\epsilon^{-1}))$, $K = \mathcal{O}(\log(\epsilon^{-2}))$, $\alpha = \mathcal{O}(T^{-1/2})$, and $T = \widetilde{\mathcal{O}}(\epsilon^{-4})$. As a result, the sample complexity and the gradient complexity over $g$ is of order $\mathcal{O}(2^K T) = \widetilde{\mathcal{O}}(\epsilon^{-6})$. At each upper iteration, we compute one gradient of $f$, and thus the gradient complexity over $f$ is $T = \widetilde{\mathcal{O}}(\epsilon^{-4})$. $\qquad\square$

# B  Computing $\nabla_1 y^*(x; \xi)$

To derive an explicit formula for $\nabla_1 y^*(x; \xi)$, we use the first-order optimality condition of the unconstrained lower-level problem. Indeed, as $g(x, y; \eta, \xi)$ is strongly convex in $y$, for given $x$ and $\xi$, $y^*(x; \xi)$ is the unique solution of the equation

$$\mathbb{E}_{\mathbb{P}_{\eta|\xi}}\nabla_2 g(x, y^*(x; \xi); \eta, \xi) = 0.$$

Taking gradients with respect to $x$ on both sides and using the chain rule, we obtain

$$\mathbb{E}_{\mathbb{P}_{\eta|\xi}}\left[\nabla_{21}^2 g(x, y^*(x; \xi); \eta, \xi) + \nabla_{22}^2 g(x, y^*(x; \xi); \eta, \xi)\nabla_1 y^*(x; \xi)\right] = 0.$$

Since $g(x, y; \eta, \xi)$ is $\mu_g$-strongly convex in $y$ for any given $x, \eta$, and $\xi$, the expected Hessian matrix $\mathbb{E}_{\mathbb{P}_{\eta|\xi}}\nabla_{22}^2 g(x, y; \eta, \xi) \in \mathbb{S}_+^{d_y}$ is invertible, and thus we find

$$\nabla_1 y^*(x; \xi)^\top = -\left(\mathbb{E}_{\mathbb{P}_{\eta|\xi}}\nabla_{12}^2 g(x, y^*(x; \xi); \eta, \xi)\right)\left(\mathbb{E}_{\mathbb{P}_{\eta|\xi}}\nabla_{22}^2 g(x, y^*(x; \xi); \eta, \xi)\right)^{-1},$$

where $\mathbb{E}_{\mathbb{P}_{\eta|\xi}}\nabla_{12}^2 g(x, y^*(x; \xi); \eta, \xi)$ is in fact the transpose of $\mathbb{E}_{\mathbb{P}_{\eta|\xi}}\nabla_{21}^2 g(x, y^*(x; \xi); \eta, \xi)$.

## C  Efficient Implementation for Hessian Vector Products

Our goal is to compute $\widehat{c}(x, y; \xi) = \nabla_{12}^2 g(x, y; \eta', \xi)\widehat{\Lambda}(x, y; \xi)\nabla_2 f(x, y; \eta'', \xi)$ in (8) efficiently.

---

**Algorithm 4** Hessian Vector Implementation for Computing $\widehat{c}(x, y; \xi)$.

---

**Input:** Iteration points $(x, y)$, samples $\xi$ and $\eta'$, $\{\eta_n\}_{n=1}^{\widehat{N}}, \eta'' \sim \mathbb{P}_{\eta|\xi}, \widehat{N} \geq 1$.
1: Compute $r_0 = \nabla_2 f(x, y; \eta'', \xi)$
2: **for** $n = 1$ to $\widehat{N}$ **do**
3:   Get Hessian-vector product $v_n = \nabla_{22}^2 g(x, y; \eta_n, \xi)r_{n-1}$ via automatic differentiation on $y$
4:   Update $r_n = r_{n-1} - \frac{1}{L_{g,1}}v_n$
5: **end for**
6: Update $r'_{N_3} = \frac{N}{L_{g,1}}r_{N_3}$
7: Get Hessian-vector product $c = \nabla_{12}^2 g(x, y; \eta', \xi)r'_{\widehat{N}}$ via automatic differentiation on $y$
**Output:** $c$
   **Remark:** Step 4 and 9 can be implemented at cost $\mathcal{O}(d)$ by as we compute $\nabla_2 g(x, y; \eta_n, \xi)r_{n-1}$ first and then differentiate over $y$ to avoid calculating Hessian matrices directly.

---

## D  Implementation Details

In this section, we provide all relevant implementation details. We fine-tune the stepsize for all approaches using the following strategy: we pick a fixed breakpoint denoted as $t_0$ and adjust the stepsize accordingly. Specifically, for the $t$-th outer iteration when $t \leq t_0$, the stepsize is set as $1/\sqrt{t}$, while for iterations beyond $t_0$, the stepsize is set as $1/t$. We choose the breakpoint $t_0$ such that training loss remains relatively stable as the number of outer iterations approaches $t_0$.

### D.1  Meta-Learning

Let $D = (a, b)$ denote the training or testing data, where $a \in \mathbb{R}^d$ represents the feature vector and $b \in [C]$ represents the label with $C$ categories. In this section, the loss $\ell_i(x, D)$ is defined as a multi-class logistic loss, given by:

$$\mathcal{L}(x; D) = -\boldsymbol{b}^T x^\top a + \log\left(1^\top e^{x^\top a}\right),$$

where the parameter $x \in \mathbb{R}^{d \times C}$ stands for the linear classifier, $\boldsymbol{b} \in \{0, 1\}^C$ stands for the corresponding one-hot vector of the label $b$. The experiment utilizes the tinyImageNet dataset, which consists of 100,000 images belonging to 200 classes. After data pre-processing, each image has a dimension of 512. For dataset splitting, we divide the dataset such that 90% of the images from each class are assigned to the training set, while the remaining 10% belong to the testing set. The meta-learning task comprises 20 tasks, with each task involving the classification of 10 classes. Additionally, we set the hyper-parameter $\lambda = 2$.

Finally, we provide additional experiments for meta-learning in Figure 4. In the left plot, we examine the performance of MAML by varying the stepsize of inner-level gradient update from

$\{5e\text{-}3, 1e\text{-}2, 5e\text{-}2, 1e\text{-}1, 2e\text{-}1\}$. From the plot we can tell that when using small stepsize, MAML tends to have similar performance whereas the performance of MAML tend to degrade when using large stepsize. In the right plot, we examine the performance of $m$-step MAML against upper-level iterations. The $m$-step MAML is an approximation of problem (4) via replacing $y_i(x)$ with the $m$-step gradient update $\widehat{y}_{i,m}(x)$, which is defined recursively:

$$\widehat{y}_{i,0}(x) = x, \quad \widehat{y}_{i,k}(x) = \widehat{y}_{i,k-1}(x) - \gamma\nabla\left[l_i(\widehat{y}_{i,k-1}(x), D_{i,k-1}^{train}) + \frac{\lambda}{2}\|\widehat{y}_{i,k-1}(x) - x\|^2\right],$$

with stepsize $\gamma$ and $D_{i,k-1}^{train} \sim \rho_i$ for $k = 1, \ldots, m$. Here we take the number of gradient updates at inner level $m$ from $\{1, 4, 8, 12\}$. From the plot, we realize that multi-step MAML tends to have better performance, but it still cannot outperform the RT-MLMC method.

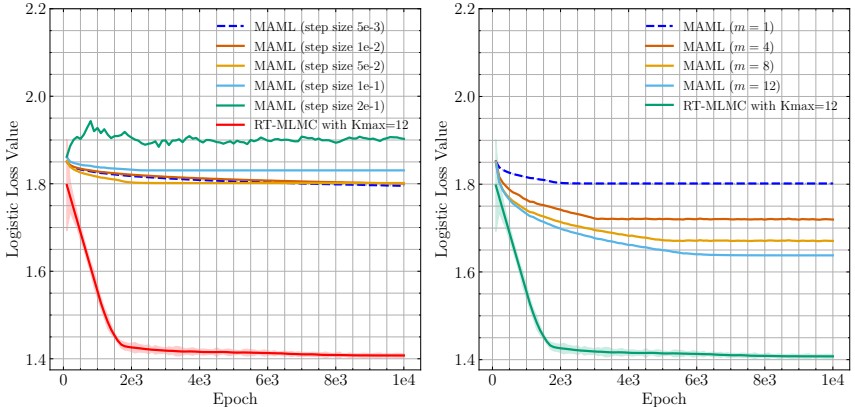

Figure 4: Left: Performance of one-step MAML against upper-level iterations on meta-learning. Right: Performance of $m$-step MAML against upper-level iterations on meta-learning.

### D.2 Wasserstein DRO with Side Information

In this subsection, we take the loss function

$$l(w, \eta) = h(w - \eta)_+ + b(\eta - w)_+,$$

where $h > 0$ is a constant representing the per-unit holding cost and $b > 0$ is a constant representing the per-unit backlog cost. Since Assumption 2.1 does not hold for the objective function due to its non-smooth structure, we approximate the loss function with the smoothed version

$$l_\beta(w, \eta) = \frac{h}{\beta}\log(1 + e^{\beta(w-\eta)}) + \frac{b}{\beta}\log(1 + e^{\beta(\eta-w)}),$$

where we specify the hyper-parameter $\beta = 5$ to balance the trade-off between loss function approximation and smoothness. The synthetic dataset in this part is generated in the following procedure: the covariate $\xi$ is sampled from the 100-dimensional uniform distribution supported on $[-15, 15]^{100}$. The demand $\eta$ depends on $\xi$ in a nonlinear way:

$$\eta = f_{\text{NN}}(x; \xi) + \epsilon, \quad \epsilon \sim \mathcal{N}(0, 1),$$
$$f_{\text{NN}}(x; \xi) = 10 * \text{Sigmoid}(W_3 \cdot \text{ReLU}(W_2 \cdot \text{ReLU}(W_1\xi + b_1) + b_2) + b_3),$$

where the neural network parameter $x := (W_1, b_1, \ldots, W_3, b_3)$. In particular, the ground-truth neural network parameter is generated using the uniform initialization procedure by calling `torch.nn.init.uniform_` in pytorch. We quantify the performance of a given $\theta$ using the testing loss $\mathbb{E}_{(\xi,\eta)\sim\mathbb{P}^*}[l(f(x;\xi), \eta)]$, where the expectation is estimated using sample average approximation based on $2 \cdot 10^5$ sample points. When creating training dataset, we generate $M = 50$ samples of $\xi$, denoted as $\{\xi_i\}_{i\in[M]}$ and for each $x_i$, we generate $m \in \{10, 30, 50, 100\}$ samples of $\eta$ from the conditional distribution $\mathbb{P}_{\eta|\xi_i}$. When generating the left two plots of Fig. 3, we specify $m = 100$.

When solving the WDRO baseline, we consider the formulation

$$\min_x \max_{\mathbb{P}} \left\{\mathbb{E}_{(\xi,\eta)\sim\mathbb{P}}[l_\beta(f(x;\xi); \eta)] - \lambda\mathcal{W}(\mathbb{P}, \mathbb{P}^0)\right\}, \tag{13}$$

with $\mathcal{W}(\cdot, \cdot)$ being the standard Wasserstein distance using the same cost function as the casual transport distance. We apply the SGD algorithm developed in [Sinha et al., 2018] to solve the WDRO formulation. See Algorithm 5 for detailed implementation. The hyper-parameter $\lambda$ for WDRO-SI or WDRO formulation has been fine-tuned via grid search from the set $\{1, 10, 50, 100, 150\}$ for optimal performance.

---

**Algorithm 5** Gradient Descent Ascent Heuristic for Solving (13)

---

**Input:** $\sharp$ of outer iterations $T$ and $\sharp$ of inner iterations $T_{\text{in}}$, stepsizes $\{\alpha_t\}_{t=1}^T$ and $\{\beta_s\}_{s=1}^{T_{\text{in}}}$, initial iterate $x_1$.

1: **for** $t = 1$ to $T$ **do**
2:     Sample $(\xi^t, \eta^t)$ from $\mathbb{P}$ and initialize $\xi_1^t = \xi^t$.
3:     **for** $s = 1$ to $T_{\text{in}}$ **do**
4:        Generate $g_s = \nabla_\xi [l_\beta(f(x_t; \xi_s^t); \eta^t) - \lambda \|\xi_s^t - \xi^t\|^2]$.
5:        Update $\xi_{s+1}^t = \xi_s^t + \beta_s \text{Adam}(\xi_s^t, g_s)$.
6:     **end for**
7:     Compute $G_t = \nabla_x [l_\beta(f(x_t; \xi_{T_{\text{in}}+1}^t); \eta^t)]$.
8:     Update $x_{t+1} = x_t - \alpha_t G_t$.
9: **end for**

**Output:** $x_T$

---

