# OpenReview forum: "Contextual Stochastic Bilevel Optimization"
_NeurIPS.cc/2023/Conference — NeurIPS 2023 poster_

### Official Review · Reviewer_DnyS · 2023-07-02

**Soundness:** 3 good
**Presentation:** 3 good
**Contribution:** 2 fair
**Rating:** 5
**Confidence:** 4

**Summary:**

This work studies the so-called contextual stochastic bilevel optimization (CSBO) problem, in which the lower-level problem is a conditional expectation problem under some contextual information. Some applications in distributional robust optimization fall into this category. The paper proposes a double-loop Monte-Carlo based method, which leverage EpochSGD in Hazan and Kale, 2014 to approximate  the lower-level solution. A approximate hypergradient is provided based on an explicit form via implicit function theorem. Experiments on meta-learning and instrumental variable regression are provided.

**Strengths:**

1. This work considers a problem that is different from existing studies by introducing the contextual information. It covers some important cases in, e.g., DRO.

2. The algorithms are reasonable to me. Using MLMC to further improve the performance of DL-SGD is good. Theoretical complexity and convergence are analyzed. Experiments seem to support the design principles.

**Weaknesses:**

1. The studied problem seems a little bit artificial. The example in (3) seems to artificially change a single-level problem into a bilevel one. In addition, the hyper-gradient form is almost the same as in the non-contextual case and hence the tools therein may be used with some adaptations.

2. Some important components such as MLMC, EpochSGD, Neumann series expansion are existing techniques. Some challenges like variance control (e.g., in $\hat v$) and hyper-gradient computation can be well coped with by these techniques. Thus, the novelty is not that significant.

3. In experiments, in Fig. 1, why does MAML stop at this large loss value? Is it because of a non-satisfactory hyperparameter tuning or something else. It seems that the stepsizes for MAML may be chosen to be too large. The comparison seems not fair.


4. The experiments do not show the important of bilevel optimization. For example, no baselines other than the proposed CSBO solvers are provided. Some baselines (e.g., single-level ones, or some standard baselines) in Wasserstein DRO may need to be included.

Overall, I am not fully convinced that this is an important problem, and given the above concerns, I lean toward the negative side.

**Questions:**

See Weakness part.

**Limitations:**

See Weakness part.

---

> ### Author Rebuttal · Authors · 2023-08-08
>
> Thank you for the comments. In below, we address your concerns.
> 1. [Motivation of CSBO] The CSBO problem is not artificially made up. In addition to optimization-based meta-learning and Wasserstein DRO with side information (also known as the causal optimal transport) mentioned in the numerical study, there are other applications that are special cases of the CSBO problems but are not special cases of Equation (3).
>     - Personalized federated learning [Xing, Pengwei, et al. "Big-fed: Bilevel optimization enhanced graph-aided federated learning." *IEEE Transactions on Big Data* (2022)] is a special case of CSBO, see Equation (2) in the reference. It is important to note that in personalized federated learning, the number of lower-level constraints can be on the order of $O(10^9)$ as each constraint represents one person. Therefore, the RT-MLMC method, which has convergence independent of the number of lower-level constraints, is crucial for achieving efficiency. Convergence of existing methods in [Guo et al., 2021] and [Hu et al., 2023 Blockwise stochastic variance-reduced methods with parallel speedup for multi-block bilevel optimization. arXiv preprint arXiv:2305.18730, 2023] all depend linearly on $M$.
>     - End-to-end learning/Contextual Optimization [Sadana, Utsav, et al. "A Survey of Contextual Optimization Methods for Decision Making under Uncertainty." *arXiv preprint arXiv:2306.10374* (2023).] In this survey, there are three paradigms for studying end-to-end learning/contextual optimization. The third paradigm, which integrates learning and optimization, falls as a special case of CSBO problems (see Figure 3 in Sadana et al., 2023). Also, see Table 3 on Page 19 in in Sadana et al., 2023 for a list of more than 20 papers that fall into this category. Our work is the first to provide non-asymptotic optimal complexity bounds for such a problem.
>
>     Indeed, we find that contextual stochastic bilevel optimization has various applications in machine learning and optimization.
>
> 2. [Novelty] Note that the contextual information $\xi$ introduces additional challenges in designing an efficient optimal algorithm. In particular, most of the existing single-loop algorithms for classical bilevel optimization do not apply to our problem due to potentially infinitely many lower-level problems parametrized by $\xi$, each of which introduces a constraint involving solving a stochastic optimization problem. Additionally, we have shown that the double-loop algorithm DL-SGD is not optimal and have proposed an optimal algorithm RT-MLMC. The closest work to ours [Guo et al., 2021] can only handle a finite number of lower-level constraints ($M$), and the complexity bounds depend linearly on $M$. Our algorithm does not depend on $M$ and sheds light on how to solve stochastic bilevel optimization with even infinitely many lower-level constraints.
> 3. [Comparison to MAML] We respectfully point out that the step size of MAML has been fine-tuned in our experiment. In our general response (see newly added Figure 4 in the PDF file), we report the plot of MAML performance for different choices of step size from the list {5e-3, 1e-2, 5e-2, 1e-1, 2e-1}. From the plot we realize that for small step sizes the MAML tends to have similar performance, whereas MAML tends to diverge for too large step sizes. The poor performance of MAML on meta-learning problem (8) is that it solves a different formulation, i.e., it replaces the lower level problems with one-step gradient update. It is natural that MAML cannot achieve good performance on the CSBO objective, as the approximation gap is theoretically $O(1)$ unless one performs multi-step MAML.
>
>     - Besides, in our general response, we provide the performance of multi-step step MAML, which replaces the lower-level problems in (8) with $m$-step gradient updates, with $m\in$ {$1,4,8,12$}. From the plot we can see as $m$ increases, multi-step MAML tends to have better performance, but it still cannot outperform our proposed RT-MLMC algorithm.
>
> 4. [Experiments against baselines] Thank you for the suggestions. As we are the first to propose and solve the CSBO problem, we have not found any other baselines for CSBO.
>
>     - Regarding Wasserstein DRO with side information, it is worth noting that the existing method in [Yang et al., 2022] heavily relies on convexity, linearity predictors, and linear loss assumptions to build reformulations that can be solved via convex solvers. However, when the hypothesis class from the covariate to the decision is parameterized by nonconvex neural networks, their method is not implementable. In contrast, our algorithm is the first implementable algorithm with a convergence guarantee. We compare our method to naively incorporating SAA and Wasserstein DRO methods that do not explicitly leverage side information in Figure 2 (right).
>
>     - For multi-task stochastic bilevel optimization [Guo et al., 2021], which is a special case of CSBO with $M$ lower-level problems, we compare the performance of RT-MLMC with the state-of-the-art method BSVRB proposed in [Hu et al., 2023] (this manuscript appears on arXiv after the NeurIPS submission deadline) in the one-page PDF response. Please see Figure 5 for a comparison with BSVRB. Our RT-MLMC method converges much faster.
>
> Xing, Pengwei, et al. "Big-fed: Bilevel optimization enhanced graph-aided federated learning." *IEEE Transactions on Big Data* (2022)
>
> Sadana, Utsav, et al. "A Survey of Contextual Optimization Methods for Decision Making under Uncertainty." arXiv preprint arXiv:2306.10374 (2023).
>
> Hu et al. "Blockwise stochastic variance-reduced methods with parallel speedup for multi-block bilevel optimization". arXiv preprint arXiv:2305.18730, 2023

---

> > ### Comment · Reviewer_DnyS · 2023-08-17
> > **Thanks for the response**
> >
> > Dear Authors,
> >
> > Thanks so much for the response! My questions have been answered satisfactorily, so I increase my score.
> >
> > Best,
> > Reviewer

---

> > > ### Author Response · Authors · 2023-08-21
> > > **Thank you for the discussion**
> > >
> > > Thank you for the time and the valuable feedback.
> > >
> > > Best Regards,
> > >
> > > Authors

---

> ### Comment · Area_Chair_gNhP · 2023-08-12
>
> Dear reviewer,
>
> Thank you for your review! The authors have replied to your comments. Does their answer address your concern? Can you please react to their answer during the discussion period?
>
> Many thanks,
>
> The AC

---

> ### Author Response · Authors · 2023-08-14
> **Follow-up on Rebuttal: Seeking Your Feedback**
>
> Dear Reviewer Dnys,
>
> We appreciate the time you've taken to review our work. We've addressed your concerns in our rebuttal, and we kindly ask if you've had an opportunity to go through it. Please let us know if there are any further questions or clarifications you would like. Thank you.
>
> Best regards,
> Authors

---

> > ### Comment · Area_Chair_gNhP · 2023-08-17
> > **Need feedback to the authors**
> >
> > Dear reviewer,
> >
> > Thank you for your review.
> > The authors provided an reply to address your concerns.
> > Could you please acknowledge that you read the response and, in case you maintain your score, to provide more details about why the reply does not address your concern.
> >
> > Thank you for your time and effort.
> >
> > Best,
> > The AC

---

### Official Review · Reviewer_NrrF · 2023-07-04

**Soundness:** 2 fair
**Presentation:** 1 poor
**Contribution:** 2 fair
**Rating:** 5
**Confidence:** 2

**Summary:**

Contextual stochastic bilevel optimization (CSBO) is introduced in this paper. An efficient double-loop gradient method based on the Multilevel Monte-Carlo (MLMC) is proposed. The proposed framework captures important applications such as meta-learning, Wasserstein distributionally robust optimization with side information (WDRO-SI), and instrumental variable regression (IV).

**Strengths:**

An interesting problem, that is, Contextual Stochastic Bilevel Optimization Problem is proposed in this work. And an efficient algorithm is proposed to solve this problem. The proposed framework captures important applications such as meta-learning, Wasserstein distributionally robust optimization with side information (WDRO-SI), and instrumental variable regression (IV). However, I have some concerns as follows.

**Weaknesses:**

1. I think it's necessary to emphasize the difficulty of solving the Contextual Stochastic Bilevel Optimization Problem compared with traditional bilevel optimization problems. The presentation of this work is poor. I believe this is an excellent work, I suggest that you should modify the presentation of this work to better clarify the contributions of this work.

2. In the experiment, the results are limited. For example, in the meta-learning application, I suggest the authors compare the proposed method with the state-of-the-art bilevel optimization methods [1][2][3], which are shown to be able to address the meta-learning task. Alternatively, you need to specify why these methods [1][2][3] are not applicable to this application. Furthermore, the authors should conduct experiments on more datasets to better evaluate the proposed method, for example, Omniglot dataset.

3. I suggest the author briefly introduce some existing bilevel optimization works in machine learning, for example, hyper-gradient-based methods [1, 4] and approximation-based methods [2], and then discuss why these methods fail to be applied to the Contextual Stochastic Bilevel Optimization Problem.

[1] Bilevel optimization: Convergence analysis and enhanced design, ICML, 2021

[2] Asynchronous Distributed Bilevel Optimization, ICLR 2023

[3] Bilevel Programming for Hyperparameter Optimization and Meta-Learning, ICML 2018

[4] Provably faster algorithms for bilevel optimization, NeurIPS 2021

**Questions:**

1. I think it's necessary to emphasize the difficulty of solving the Contextual Stochastic Bilevel Optimization Problem compared with traditional bilevel optimization problems. The presentation of this work is poor. I believe this is an excellent work, I suggest that you should modify the presentation of this work to better clarify the contributions of this work.

2. In the experiment, the results are limited. For example, in the meta-learning application, I suggest the authors compare the proposed method with the state-of-the-art bilevel optimization methods [1][2][3], which are shown to be able to address the meta-learning task. Alternatively, you need to specify why these methods [1][2][3] are not applicable to this application. Furthermore, the authors should conduct experiments on more datasets to better evaluate the proposed method, for example, Omniglot dataset.

3. I suggest the author briefly introduce some existing bilevel optimization works in machine learning, for example, hyper-gradient-based methods [1, 4] and approximation-based methods [2], and then discuss why these methods fail to be applied to the Contextual Stochastic Bilevel Optimization Problem.

[1] Bilevel optimization: Convergence analysis and enhanced design, ICML, 2021

[2] Asynchronous Distributed Bilevel Optimization, ICLR 2023

[3] Bilevel Programming for Hyperparameter Optimization and Meta-Learning, ICML 2018

[4] Provably faster algorithms for bilevel optimization, NeurIPS 2021

---

> ### Author Rebuttal · Authors · 2023-08-08
>
> Thank you for your comments. Below, we address your questions.
> 1. [Difficulty for solving CSBO] As discussed in the Introduction from Line 43 to Line 68, existing algorithms for traditional bilevel optimization problems with one single lower-level problem either cannot achieve optimal complexity bounds or does not apply to the CSBO problem due to potentially infinitely many lower-level problems parametrized by $\xi$, each of which introduces a constraint involving solving a stochastic optimization problem. See more details in below. We highlight that the proposed RT-MLMC achieves the optimal complexity bounds for CSBO problems. It is also the first double-loop algorithm that achieves the optimal complexity bounds for classical stochastic bilevel optimization.
>     - Existing double-loop methods for classical bilevel optimization can be applied with a slight modification but admits a sub-optimal complexity bound, as we have shown in DL-SGD.
>     - As for the more widely investigated single-loop methods for classical bilevel optimization, these algorithms are specifically designed for the setting when there is only one lower-level constraint formulated as $y^*(x)$. In such a case, one can use a sequence of vectors $y^t$ to approximate $y^*(x^t)$ leveraging the intuition that $y^*(x^{t+1}) - y^*(x^t) \approx \nabla y^*(x^t) (x^{t+1} - x^t)$. However, when there are multiple lower-level constraints, i.e., $y^*(x;\xi)$ for each realization of $\xi$, we cannot no longer use only one sequence of $y^t$ to keep track of $y^*(x^t,\xi^t)$ as $\xi^t$ is randomly sampled at each iteration. For example, when $\xi$ follows a normal distribution, it requires infinite number of sequence of $y$ to keep track of the lower level constraints.
>    - To our best knowledge, [Guo et al., 2021] is the only work that discussed the case when there are $M$ lower-level constraints, and they adopt $M$ sequences of $\{y^t(i)\}$ to keep track of $y^*(x;i)$  for $i\in[M]$, respectively. Thus, the convergence rate of their algorithm is depends linearly on $M$. On the other hand, our proposed RT-MLMC algorithm gets rid of the dependence on the number of lower-level constraints and achieves an optimal complexity bound of $O(\epsilon^{-4})$. This is particularly important for many applications that are special cases of the CSBO, including personalized federated learning [Xing, et al. "Big-fed: Bilevel optimization enhanced graph-aided federated learning." *IEEE Transactions on Big Data* (2022)]. In this context, $M$ can be of size $O(10^9)$, meaning that each lower-level constraint represents a personalized keyboard usage preference.
> 2. [Comparison to baselines in meta-learning] We respectfully point out that the references mentioned [1-4] address the traditional bilevel optimization problem, where there is only one lower-level constraint. When applied to meta-learning applications, they do not treat all tasks as multiple individual lower-level constraints. Instead, they use a surrogate aggregated lower-level objective to enforce one lower-level problem, for example, via averaging. In other words, they do not solve the bilevel optimization formulation for meta-learning proposed in [Rajeswaran et al., 2019] and studied in our paper. It is unclear how the proposed methods in [1-4] can be applied to the meta-learning formulation studied in our paper. Note that the meta-learning formulation in our paper is also used in personalized federated learning [Xing, Pengwei, et al., 2022]. If one adapts the surrogate aggregated lower-level constraint, it is not personalized at all. It implies that importance of solving the formulation (8) considered in our paper. Yet no algorithms in [1-4] can be applied.
>     - In the general response (also see the added PDF file), we have added comparison to two recent papers for solving a special of CSBO when there are $M$ lower-level problems [Guo and Yang, 2021] and [Hu et al. 2023, Blockwise stochastic variance-reduced methods with parallel speedup for multi-block bilevel optimization. arXiv preprint arXiv:2305.18730, 2023] (the second paper appeared on ArXiv after the NeurIPS submission deadline). The algorithm in [Guo and Yang, 2021] and the first algorithm in [Hu et al. 2023], both require computing the inverse of Hessian exactly in each iteration, which cannot be implemented efficiently especially for high-dimensional problems. We use the BSVRB, the second algorithm in [Hu et al. 2023], as a baseline to solve the special case of CSBO formulation. The comparison is in the general response with supporting Figure 5 in the PDF file. Note that the performance of BSVRB is worse than RT-MLMC. The reason is that the iteration complexity of the baseline BSVRB depends linearly on the number of lower-level problems, whereas the proposed RT-MLMC does not and achieves optimal complexity bounds.
>    -  We note that meta-learning with Omniglot dataset is a high-dimensional bilevel optimization problem with tremendously many lower-level problems, which is difficult to solve given the short rebuttal time period. We promise to add experiments either using this dataset or other types of datasets such as UCI Adult benchmark dataset *a8a* and web page classification dataset *w8a* in our revised paper.
>
> 3. [More reference] Thanks for the reference. We have added a related discussion. The reasons why these algorithms fail for CSBO are discussed in the first bullet point.
>
> Xing et al. "Big-fed: Bilevel optimization enhanced graph-aided federated learning." IEEE Transactions on Big Data (2022).
>
> Hu et al. "Blockwise stochastic variance-reduced methods with parallel speedup for multi-block bilevel optimization". arXiv preprint arXiv:2305.18730, 2023

---

> > ### Comment · Reviewer_NrrF · 2023-08-21
> >
> > Thanks for your responses, my concerns have been addressed, and I have increased the score.

---

> > > ### Author Response · Authors · 2023-08-21
> > > **Thank you for the discussion**
> > >
> > > Thank you for the time and the valuable feedback.
> > >
> > > Best Regards,
> > >
> > > Authors

---

> ### Comment · Area_Chair_gNhP · 2023-08-12
>
> Dear reviewer,
>
> Thank you for your review! The authors have replied to your comments. Does their answer address your concern? Can you please react to their answer during the discussion period?
>
> Many thanks,
>
> The AC

---

> ### Author Response · Authors · 2023-08-14
> **Follow-up on Rebuttal: Seeking Your Feedback**
>
> Dear Reviewer NrrF,
>
> We appreciate the time you've taken to review our work. We've addressed your concerns in our rebuttal, and we kindly ask if you've had an opportunity to go through it. Please let us know if there are any further questions or clarifications you would like. Thank you.
>
> Best regards,
>
> Authors

---

> > ### Comment · Area_Chair_gNhP · 2023-08-17
> > **Need feedback to the authors**
> >
> > Dear reviewer,
> >
> > Thank you for your review.
> > The authors provided an reply to address your concerns.
> > Could you please acknowledge that you read the response and, in case you maintain your score, to provide more details about why the reply does not address your concern.
> >
> > Thank you for your time and effort.
> >
> > Best,
> > The AC

---

### Official Review · Reviewer_Z7Go · 2023-07-06

**Soundness:** 3 good
**Presentation:** 3 good
**Contribution:** 3 good
**Rating:** 6
**Confidence:** 4

**Summary:**

**Summary:** The paper introduces a novel algorithm for solving contextual stochastic bilevel optimization (CSBO) problems. The authors develop DL-SGD and RT-MLMC gradient estimators for the problem along with a SGD-based algorithm for solving the CSBO problem. The authors analyze the properties of the gradient estimators and provide finite-time convergence guarantees for the proposed method. The theoretical results are corroborated by experiments on MAML, DRO with side information, and IV regression problems.

**Strengths:**

**Strengths:**

-	The problem formulation considered in the paper is challenging and is of significant interest to the ML community.
-	The paper is well written, and the ideas are presented clearly with discussions.
-	The authors provide theoretical guarantees for the proposed approaches with an analysis of the proposed gradient estimators.
-	The authors have conducted experiments on multiple tasks including MAML, DRO with side information, and IV regression problem to evaluate the performance of the proposed framework.


**Weaknesses:**

**Weaknesses:**

1. A major confusion I have is about the unbiasedness of the gradient estimator stated after line 127. Specifically, the expressions $\nabla_1 y^\ast$ and $\nabla_2 f$ both depend on the random variable $\xi$. This implies that the two expressions are dependent on each other which further means that the expectation given in the equation (after line 127) will not be the same as $\nabla F$. Can the authors clarify why the given expression will be true? The same discussion holds for the approximate gradient expressions derived in eq (5) and the rest of the paper. This is a major issue since the proofs and the results are based on the independence of $\nabla_1 y^\ast$ and $\nabla_2 f$. \
          If this issue is resolved, I am willing to raise my score.

2. The authors should clarify the intuition behind using EpochSGD to estimate $y^\ast$. In the discussion, the authors simply explain the algorithm without explaining the intuition.
3. The gradient inverse estimator utilized from Ghadhimi’s work is clear, however, why the iterative estimator RT-MLMC works is difficult to understand. Can the authors please explain the working of the proposed estimator, i.e., why it works?
4. The experiments on MAML and DRO do not compare the proposed approach with baseline algorithms in the area.

    - Numerous bilevel algorithms solve MAML, the authors should compare the performance of the proposed approach against at least a few of them.

    - Similarly, the authors need to show the performance of the proposed approach for solving the DRO problem against popular baselines.

5. It would be easier if the authors keep the notations consistent in the experiment and the theory section of the paper.

6. To motivate the problem better the authors should include some examples in the introduction section. This way it will be easier for the reader to appreciate the considered formulation.

----
Updated the score after the rebuttal.


**Questions:**

Please see the limitations above.

---

> ### Author Rebuttal · Authors · 2023-08-08
>
> Thanks for your comments. Below, we address your comments in detail.
>
> 1. [Form of gradient estimator $\nabla F(x)$] Indeed $y^*(x;\xi)$ and $\nabla_2 f(x,y^*(x;\xi);\eta,\xi)$ both depend on $\xi$. However, this does not prevent us from obtaining the expression for the gradient. Note that:
> $$F(x)=\mathbb{E}[f(x,y^*(x;\xi);\eta,\xi)]=\mathbb{E}\_\xi\mathbb{E}\_{\eta\mid\xi}[f(x,y^*(x;\xi);\eta,\xi)].$$
> By the fact that one can switch expectation and taking gradient under given assumptions, we have:
> $$\nabla F(x)=\mathbb{E}\_\xi\nabla_x[\mathbb{E}\_{\eta\mid\xi}[f(x,y^*(x;\xi);\eta,\xi)]],$$
> Note that for a given $\xi$, the gradient formulation follows exactly from that of the classical stochastic bilevel optimization without dependence structure using chain rule. See, for instance, [Ghadimi and Wang, 2018] for more details.
> $$\nabla_x[\mathbb{E}\_{\eta\mid\xi}[f(x,y^*(x;\xi);\eta,\xi)]]=  \mathbb{E}\_{\eta\mid\xi}[\nabla_x  f(x,y^*(x;\xi);\eta,\xi)]=\mathbb{E}\_{\eta\mid\xi}[\nabla_1 f(x,y^*(x;\xi);\eta,\xi) + \nabla_1 y^*(x;\xi)^\top \nabla_2 f(x,y^*(x;\xi);\eta,\xi)].$$
> Further taking expectation with respect to $\xi$ gives the expression of  $\nabla F(x)$ after Line 127, i.e.,
> $$\nabla F(x)=\mathbb{E}\_\xi\mathbb{E}\_{\eta\mid\xi}[\nabla_1f(x,y^*(x;\xi);\eta,\xi)+\nabla_1 y^*(x;\xi)^\top\nabla_2 f(x,y^*(x;\xi);\eta,\xi)].$$
> Plugging in $\nabla_1y^*(x;\xi)$ obtained in Appendix B, we have the expression of $\nabla F(x)$ after Line 129.
> 2. [Intuition to use EpochSGD] There are two reasons for using EpochSGD instead of classical SGD. Firstly, EpochSGD is faster than SGD by a logarithmic factor and is optimal in terms of sample and iteration complexity (see Hazan and Kale, 2014). Additionally, MLMC-based methods use the control variate technique between two neighboring approximations. If the difference between two neighboring approximations is too small, the approximation error decays too slowly. If the difference is too large, the variance reduction effect achieved by the control variate is not strong enough. To achieve a balance, MLMC usually uses a sequence of approximations that admits exponentially decaying approximation error. In this regard, the error of EpochSGD decays exactly exponentially in the number of epochs. Therefore, we can use the output at the end of each epoch for MLMC and do not need to manually select at which iteration of classical SGD we use the corresponding $y$ in the construction of MLMC (see Asi et al., 2021 for more discussion on EpochSGD).
> 3. [Motivation of RT-MLMC estimator] Although DL-SGD is low-biased, the complexity of DL-SGD is too high. Thus we want to construct an unbiased estimator of DL-SGD at an even lower cost to achieve optimal complexity bounds. Note that even for stochastic bilevel optimization, existing double-loop methods cannot achieve optimal complexity.
>     - To ensure low bias, we use the equation after Line 163 and show that the RT-MLMC gradient estimator in Eq. (6) to be an unbiased estimator of DL-SGD in Eq.(5) and thus low-bias for the overall objective.
>     - To achieve low costs, RT-MLMC uses a randomized approach by incorporating a truncated geometric distribution $p_k\propto 2^{-k}$ to generate an approximation level $k$. With a high probability, RT-MLMC generates a small $k$ and constructs $\hat{v}^{k+1}$ and $\hat{v}^k$, which only requires computing $y_{k+1}^0$ and $y_{k}^0$. This is inexpensive because $k$ is very small. With a low probability, RT-MLMC generates a large $k$ and computes $\hat{v}^{k+1}$ and $\hat{v}^k$, which is very expensive. The expected computational cost is thus mild by summation over high costs multiplied by low probability + low costs multiplied by high probability.
>     - Lastly, one might argue that dividing $p_k$ for small $k$ can cause high variance. This is mitigated as RT-MLMC incorporates a control variate technique by subtracting $\hat{v}^{k+1}$ and $\hat{v}^k$ that are highly correlated.
>     - In summary, RT-MLMC is unbiased of DL-SGD, thus admitting low-bias for $\nabla F(x)$; it has much lower costs than DL-SGD and admits mild variance.
>
> 4. [Compare to baseline in experiments] Note that even for a special case of CSBO, i.e., Problem (8) that is a stochastic bilevel optimization with multiple lower-level problems, many baseline approaches in bilevel optimization literature are not applicable. For meta-learning in form of Problem (8), many existing methods actually only solve a surrogate of the formulation by replacing the lower-level problem with gradient updates to obtain one-step or multi-step MAML or by averaging all lower-level problems so that there is only one lower-level constraint. However, the original formulation actually finds a lot important applications, such as personalized federated learning. To directly solve Problem (8), we provide detailed comparisons with baseline methods including BSVRB [Hu et al., 2023 Blockwise stochastic variance-reduced methods with parallel speedup for multi-block bilevel optimization. arXiv preprint arXiv:2305.18730, 2023] (the STOA that appears even after the NeurIPS deadline) and one-step/multi-step MAML in the general response with figures in the PDF file. In short, RT-MLMC outperforms these methods under various experimental setups.
>      - For Wasserstein DRO with side-information, the existing method in [Yang et al., 2022] heavily relies on convexity, linearity predictors, and linear loss assumptions to build reformulations that can be solved by convex solvers. However, when the hypothesis class from the covariate to the decision is parameterized by nonconvex neural networks, their method is not implementable. In contrast, our algorithm is the first implementable algorithm with a convergence guarantee.
> 5. [Consistent notation in theory and experiments] We will keep the notation in the theory and the experiments consistent.
> 6. [Include examples in the Introduction] We have added the applications into the introduction for a better illustration.

---

> > ### Comment · Reviewer_Z7Go · 2023-08-16
> > **Thank you for the response**
> >
> > I thank the authors for the detailed response. Most of my concerns have been addressed by the authors, especially, the ones concerning the unbiasedness of the gradient estimator. Consequently, I have updated my original rating of the paper.

---

> > > ### Author Response · Authors · 2023-08-21
> > > **Thank you for the discussion**
> > >
> > > Thank you for the time and the valuable feedback.
> > >
> > > Best Regards,
> > >
> > > Authors

---

> ### Comment · Area_Chair_gNhP · 2023-08-12
>
> Dear reviewer,
>
>
> Thank you for your review!
> The authors have replied to your comments. Does their answer address your concern?
> Can you please react to their answer during the discussion period?
>
>
> Many thanks,
>
> The AC

---

> ### Author Response · Authors · 2023-08-14
> **Follow-up on Rebuttal: Seeking Your Feedback**
>
> Dear Reviewer Z7Go,
>
> We appreciate the time you've taken to review our work. We've addressed your concerns in our rebuttal, and we kindly ask if you've had an opportunity to go through it. Please let us know if there are any further questions or clarifications you would like. Thank you.
>
> Best regards,
> Authors

---

### Official Review · Reviewer_BuTe · 2023-07-24

**Soundness:** 3 good
**Presentation:** 4 excellent
**Contribution:** 4 excellent
**Rating:** 7
**Confidence:** 3

**Summary:**

The paper investigates a generalization of the stochastic bilevel optimization model, in which the lower and upper optimization levels share a random variable. The authors design two gradient-based approaches named RT-MLMC and DL-SGD for solving this problem and analyze their performance. The two methods differ in the way they estimate the gradient of the objective: DL-SGD is rather straightforwardly based on existing results, while RT-MLMC improves upon the performance of DL-SGD by a carefully tailored sampling method. Finally, numerical examples are presented to demonstrate the generality and performance of the proposed model and methods.


**Strengths:**

* The paper introduces a general model that is applicable in a wide range of situations. Although the model is more general than previous approaches, the proposed approximations appear to be as efficient as the methods developed for the special cases.

* The paper is clear, focused, and well-written.

* I have verified most of the math; it is easy to follow and, aside from a few typos, appears to be correct.


**Weaknesses:**

There are several typos in critical parts of the proofs (see "Questions" below). I believe that these typos do not affect the correctness of the proofs, however, this somewhat lowers my confidence in the results.


**Questions:**

1. Is the dependence by $f$ on $\eta$ in the upper level of (1) necessary? I believe that removing it does not hurt the generality of the model and may avoid some confusion (see typos below).

2. In Lemma 1, $\alpha$ should be defined.

3. Typos:
* In (5), $\eta'$ is used as the argument for the $\nabla_1 f$ and $\nabla^2_{12}g$ terms while $\eta''$ is used in the $\nabla_2 f$ term. This is in contrast with the second formula for $\nabla F$ on Page 4, where an independent RV is used in $\nabla^2_{12}g$.
* A similar typo appears on the bottom of Page 15, in the proof of Lemma 2, where $\eta$ in the term $\nabla^2_{12}g$ should be changed to $\eta'$ (as in the definition of $V(x)$ above).
* A similar typo appears on Page 16, where $H_K(1)$ and the last term in $H_K(2)$ should share the same variable $\eta$ (compare the formula for $V(x)$ on page 16 versus the definition of $V(x)$ on Page 15).

---

> ### Author Rebuttal · Authors · 2023-08-08
>
> Thank you for your comments. Below, we address your comments.
>
> 1. [Dependence on $f$ and $\eta$] Indeed, the dependence between $f$ and $\eta$ is not necessary. In many cases, we can remove $\eta$ in $f$ and remove the expectation over $\eta$ in the upper level. We are writing in the current way to be as most general as possible. Thanks for pointing it out!
> 2. [Define $\alpha$ in Lemma 1] We have fixed that.
> 3. [Typos] Thanks for checking our paper carefully. Indeed, to align with the second $\nabla F$ shown on page 4, one should use $\eta^\prime$ in $\nabla_{12} g$ and use $\eta^{\prime\prime}$ in $\nabla_1 f$ and $\nabla_2 f$ in Equation (5). We have made the modifications according to your suggestion. Note that the current Equation (5) is also valid because $\nabla_{12} g$ uses a different sample compared to $\nabla_2 f$. Since Equation (5) is of the form $\nabla_1 f(\eta^\prime) - \nabla_{12} g(\eta^\prime) \Lambda \nabla_2 f(\eta^{\prime\prime})$ (where we omit other dependence), after taking the full expectation, it still aligns with the second $\nabla F$ shown on Page 4. The key part is that we should not use the same sample of $\eta$ for $\nabla_{12} g$ and $\nabla_2 f$, which will lead to correlation issues. The other two places are typos. Thanks for pointing it out.
>
> We hope that the clarification could increase your confidence in our work.

---

> > ### Comment · Reviewer_BuTe · 2023-08-18
> > **Acknowledgment**
> >
> > I thank the authors for their responses.
> >
> > Regarding question #1, I see no loss in generality in removing $\xi$ from the upper level since we can always include a copy of $\xi$ as part of $\eta$ (i.e., set $\eta' = (\eta, \xi)$). In any case, this is a very minor point and the authors should choose the form they prefer best.

---

> > > ### Author Response · Authors · 2023-08-21
> > > **Acknowledgment**
> > >
> > > Thank you for the time and the valuable feedback.
> > >
> > > Best Regards,
> > >
> > > Authors

---

### Author Rebuttal · Authors · 2023-08-08

We are grateful to the reviewers for constructive comments and suggestions, which significantly improve the quality of our paper. We are happy to engage in further discussion. We first restate the importance of CSBO problem in applications and the challenges for solving it. You may find added experiments in the attached PDF.

1.  In addition to the ones demonstrated in the numerical experiments, CSBO covers many other important applications, including personalized federated learning [Xing, et al. "Big-fed: Bilevel optimization enhanced graph-aided federated learning." IEEE Transactions on Big Data (2022)] and end-to-end learning that integrates learning and optimization (See the third paradigm in the survey [Sadana, Utsav, et al. "A Survey of Contextual Optimization Methods for Decision Making under Uncertainty." arXiv preprint arXiv:2306.10374 (2023).])

2.  Existing algorithms for traditional bilevel optimization problems with one single lower-level problem either cannot achieve optimal complexity bounds or does not apply to the CSBO problem due to potentially infinitely many lower-level problems parametrized by $\xi$, each of which introduces a constraint involving solving a stochastic optimization problem. We highlight that the proposed RT-MLMC achieves the optimal complexity bounds for CSBO problems. We illustrate in more details in the response to each reviewer.

In below, we add some extra experiments to address some concerns from reviewers.

1. Reviwer Dnys asked about the step size of MAML in meta-learning. We point out that the step size of MAML has been fine-tuned in our experiment. In our general response (Figure 4a), we report the plot of MAML performance for different choices of step size from the list {5e-3, 1e-2, 5e-2, 1e-1, 2e-1}. From the plot we can see that for small step sizes the MAML tends to have similar performance, whereas MAML tends to diverge for too large step sizes. The key issue of MAML is that it solves a different formulation than problem (8), i.e., it replaces the lower level problems with one-step gradient update. Thus, it is natural that MAML cannot achieve good performance on the CSBO objective function, as the approximation gap is theoretically $O(1)$ unless one performs multi-step MAML.
    - To further illustrate the performance of MAML, in our general response, we provide the performance of multi-step MAML in the plot of Figure 4b, which replaces the lower level problems in (8) with $m$-step gradient updates, with $m\in \\{1,4,8,12\\}$. From the plot we can see as $m$ increases, multi-step MAML tends to have better performance, but it still cannot outperform our proposed RT-MLMC algorithm.

2. Reviewers Z7Go and NrrF pointed out that we should add more baseline comparison for solving the meta-learning formulation (8), which is a special case of CSBO problem. Note that it is a stochastic bilevel optimization problem with multiple lower-level constraints. Many baseline approaches can actually only solve a surrogate of the formulation by replacing the lower-level problem with gradient updates to obtain one-step or multi-step MAML or by averaging all lower-level problems so that there is only one lower-level constraint.
    - The performance of MAML is discussed in the previous bullet points.
   - Only two recent papers proposed algorithms for directly solving the formulation (8): [Guo and Yang, 2021] and [Hu et al. 2023] (note that [Hu et al. 2023] appears after the NeurIPS deadline. We still use it as a baseline). The algorithm in [Guo and Yang, 2021] and the first algorithm in [Hu et al. 2023], both require computing the inverse of Hessian exactly in each iteration, which cannot be implemented efficiently especially for high-dimensional problems (For instance, in our meta-learning experiment the dimension of decision variable is $512\times10=5120$, so these two algorithms will require inverting a $5120\times5120$ size matrix in each iteration). We use the $\mathrm{BSVRB}^{v2}$, the second algorithm in [Hu et al. 2023], as a baseline to solve the special case of CSBO formulation. Figure 5 in the PDF file compare the performance of $\mathrm{BSVRB}^{v2}$ and the proposed RT-MLMC. Note that the performance of $\mathrm{BSVRB}^{v2}$ is worse than RT-MLMC. The reason is that the iteration complexity of $\mathrm{BSVRB}^{v2}$  depends linearly on the number of lower-level problems, whereas the proposed RT-MLMC does not and achieves optimal complexity bounds.
3. Reviewer Z7Go asked to add more baseline methods for Wasserstein DRO with side information. Note that the existing method in [Yang et al., 2022] heavily relies on convexity, linearity predictors, and linear loss assumptions to build reformulations that can be solved via convex solvers. However, when the hypothesis class from the covariate to the decision is parameterized by nonconvex neural networks, their method is not implementable. In contrast, our algorithm is the first implementable algorithm with a convergence guarantee. We compare our method to baselines such that naively incorporating SAA and Wasserstein DRO methods that do not explicitly leverage side information in Figure 2 (c). We again summarize the numerical results in Figure 2(c) as Table 5 in our PDF file.

References:
- Zhishuai Guo and Tianbao Yang. Randomized stochastic variance-reduced methods for stochastic bilevel optimization. arXiv preprint arXiv:2105.02266, 2021
- Quanqi Hu, Zi-Hao Qiu, Zhishuai Guo, Lijun Zhang, and Tianbao Yang. Blockwise stochastic variance-reduced methods with parallel speedup for multi-block bilevel optimization. arXiv preprint arXiv:2305.18730, 2023. (Appears on ArXiv after the NeurIPS submission deadline.)

---

### Decision · Program_Chairs · 2023-09-21

**Decision:**

Accept (poster)

**Comment:**

The paper investigates a generalization of the stochastic bilevel optimization model, in which the lower and upper optimization levels share a random variable. The authors design two gradient-based approaches named RT-MLMC and DL-SGD for solving this problem and analyze their performance. Finally, numerical examples are presented to demonstrate the generality and performance of the proposed model and methods.

I agree with the reviewers' assessment that this is a good contribution that addresses a common practical problem in bilevel optimization. I encourage the authors to take into account the reviewers' feedback for the final version of the paper.